# Privacy-Preserving Federated Learning via Homomorphic Adversarial Networks

## Abstract

Privacy-preserving federated learning (PPFL) aims to train a global model for multiple clients while maintaining their data privacy. However, current PPFL protocols exhibit one or more of the following insufficiencies: considerable degradation in accuracy, the requirement for sharing keys, and cooperation during the key generation or decryption processes. As a mitigation, we develop the first protocol that utilizes neural networks to implement PPFL, as well as incorporating an Aggregatable Hybrid Encryption scheme tailored to the needs of PPFL. We name these networks as *Homomorphic Adversarial Networks* (HANs) which demonstrate that neural networks are capable of performing tasks similar to multi-key homomorphic encryption (MK-HE) while solving the problems of key distribution and collaborative decryption. Our experiments show that HANs are robust against privacy attacks. Compared with non-private federated learning, experiments conducted on multiple datasets demonstrate that HANs exhibit a negligible accuracy loss (at most 1.35%). Compared to traditional MK-HE schemes, HANs increase encryption aggregation speed by 6,075 times while incurring a 29.2-fold increase in communication overhead.

## 1 Introduction

Federated Learning (FL) has emerged as a promising paradigm for collaborative model training without direct data sharing (McMahan et al., 2017; Konečný et al., 2016). While initially believed to preserve privacy (Li et al., 2021; Yang et al., 2019), recent studies have revealed vulnerabilities in FL, demonstrating that client-side gradients can potentially leak sensitive training data (Hitaj et al., 2017; Melis et al., 2019; Zhu et al., 2019; Carlini et al., 2022).

To prevent data reconstruction in FL settings, researchers have been exploring various strategies, notably differential privacy (DP) (Wei et al., 2020; Iyengar et al., 2019; Geyer et al., 2017) and homomorphic encryption (HE) (Shi et al., 2023; Wibawa et al., 2022; Madi et al., 2021; Zhang et al., 2020b;a; Chen et al., 2019).DP stands out for its computational efficiency but may potentially reduce the performance of the FL model.

Regarding HE, although it preserves the model's performance, it may compromise the data privacy of all honest participants if a client conspires with an external attacker to share the key (collusion attacks) (Cai et al., 2023; Fang & Qian, 2021).

To mitigate this problem, Multi-Key Homomorphic Encryption (MK-HE) (Chen et al., 2019) has been proposed, which is designed to prevent collusion attacks without compromising the model's performance. However, the implementation of MK-HE introduces its own challenges, such as cooperation during the key generation or decryption processes (Park et al., 2022). These issues underscore the persistent dilemma faced in FL, how to find the right trade-off between data privacy and the practical constraints of model performance as well as resource allocation.

To address these challenges, we propose Homomorphic Adversarial Networks (HANs), a novel privacy-preserving approach that leverages neural networks to emulate the behavior of MK-HE (comparisons shown in Table 1). HANs are designed to optimize encryption and aggregation tasks without the need for traditional key distribution or collaborative decryption, thereby significantly simplifying deployment in FL scenarios. The HANs framework employs an Aggregatable Hybrid Encryption (AHE) scheme, which synthesizes the advantages of both symmetric and asymmetric

cryptography while addressing their respective limitations in the context of FL. The proposed AHE scheme introduces three cryptographic primitives: Key Generation ($KeyGen$), Encryption ($Enc$), and Aggregation ($Aggregate$), each tailored to meet the specific demands of distributed training environments.

HANs offer several key advantages over traditional privacy-preserving techniques in FL. Unlike MK-HE, HANs do not require a cumbersome key distribution process or collaborative decryption, making implementation more straightforward and practical. Additionally, HANs exhibit strong resistance to collusion attacks, even in scenarios where a majority of participants are compromised. The use of efficient One-Time Pad (OTP) and Privacy-Preserving Update (PPU) mechanisms further safeguards sensitive information, providing a robust privacy-preserving solution for FL environments.

Table 1: Comparisons of HANs with other privacy-preserving federated learning Methods

| Feature | DP | HE | MK-HE | HANs |
|---|---|---|---|---|
| Low Accuracy Loss | × | ✓ | ✓ | ✓ |
| No Key Distribution Required | ✓ | × | × | ✓ |
| No Collaborative Decryption | ✓ | ✓ | × | ✓ |
| Strong Collusion Attack Resistance | ✓ | × | ✓ | ✓ |
| Low OTP Overhead | N/A | × | × | ✓ |
| Irreversible Ciphertext | N/A | × | × | ✓ |

***Contributions.*** Our contributions are as follows.

1. We use neural networks to emulate MK-HE algorithms, enabling efficient encryption and aggregation in FL through the proposed AHE scheme. Additionally, we introduce the PPU mechanism to enhance privacy guarantees. AHE approach uses private key encryption to provide irreversible ciphertext, offering new insights into neural network-based cryptography in FL.

2. The HANs framework effectively balances privacy, performance, and efficiency by eliminating the need for collaborative decryption and key sharing. Our approach allows for the use of OTP and PPU with minimal cost while ensuring privacy, even if $N - 2$ clients collude with the server.

3. We designed a multi-stage training strategy to balance security and usability. Empirical evaluations demonstrate that HANs with AHE are practical in FL scenarios, showing only 1.35% accuracy loss compared to non-private FL, while improving encryption aggregation speed by 6,075 times, with a 29.2-fold increase in communication overhead.

## 2 RELATED WORK

### 2.1 PRIVACY-PRESERVING FEDERATED LEARNING (PPFL)

**Differential Privacy** is a frequently utilized tool for privacy protection. These studies (Abadi et al., 2016; Geyer et al., 2017; Triastcyn & Faltings, 2019; Hu et al., 2020; Kim et al., 2021; Rahman et al., 2018) have utilized DP to secure data and user privacy. However, if there is a need to prevent the reconstruction of data, the inclusion of DP can significantly compromise the accuracy of the models.

**Homomorphic Encryption** facilitates the execution of computations directly on a ciphertext to yield an encrypted outcome. Aono et al. (2017) proposed the application of HE for the safeguarding of gradient updates during the FL training procedure. Chen et al. (2020), an FL framework specifically designed for wearable healthcare, manages to achieve model aggregation by deploying HE. The application approach of this HE is expedient. Apart from encryption and decryption, it necessitates no significant alterations and imposes no extraordinary constraints on the algorithm. Importantly, the accuracy of learning is preserved with HE, as no noise infiltrates the model updates during either the encryption or decryption stages. Fang & Qian (2021) employs an enhanced Paillier algorithm to expedite computation. Zhang et al. (2020a) proposed BatchCrypt, a FL framework

based on batch encryption, with the goal of reducing computational expenses. However, traditional HE schemes require key sharing which relies on the assumption that there is no collusion between the *Server* and *Clients* (Aono et al., 2017).

In order to prevent collusion attacks, MK-HE allows multiple parties to utilize distinct keys for encryption. The decryption process necessitates the collaborative involvement of all parties. Ma et al. (2022) introduced a PPFL framework based on xMK-CKKS, demonstrating resilience against collusion involving fewer than $N$-1 participant devices and the *Server*.

However, this approach involves additional computational overhead during key generation and decryption, and requires collaboration among multiple clients for decryption. SecFed, an innovative federated learning framework, harnesses multi-key homomorphic encryption and trusted execution environments to safeguard multi-user privacy while boosting computational efficiency (Cai et al., 2023). This secure system also integrates an offline protection mechanism to address user dropout issues effectively.

## 2.2 CRYPTOGRAPHY BASED ON GENERATIVE ADVERSARIAL NETWORKS

In recent years, using neural networks, especially Generative Adversarial Networks (GANs), for encryption has become an emerging direction in cryptography research. Abadi & Andersen (2016) proposed a method to learn symmetric encryption protocols based on GANs.

They used two neural networks for encryption and decryption respectively, and introduced an attacker network to evaluate security. Subsequent works built upon this foundational research, further refining the approach (Luo et al., 2023; An et al., 2023; Li et al., 2020; Pattanayak & Ludwig, 2018). These studies introduced various enhancements, including diverse attack models and encrypted training schemes, thereby advancing the field of neural network-based cryptography.

While these works have significantly contributed to the application of neural networks in cryptography, they still face certain limitations. Primarily, they focus on message encryption without addressing homomorphic computation. Moreover, they do not adequately tackle the challenges of key distribution or negotiation, which are crucial aspects of practical cryptographic systems.

Inspired by these studies, particularly their loss function design and the application of GANs in training encryption neural networks, we propose HANs to address the aforementioned limitations and provide an enhanced solution for privacy protection within the FL context.

## 3 HANs SYSTEM DEFINITION

We proposeHANs, which leverage the AHE algorithm to meet the privacy-preserving requirements of FL. For a detailed explanation of the PPFL problem setting and system goals, please refer to Appendix A.

### 3.1 DESIGN CONCEPT OF AHE

This AHE scheme, tailored for PPFL, uses private keys for encryption and public keys for aggregation, protecting individual client data while enabling efficient aggregation without relying on trusted third parties. The key concepts of AHE are as follows:

- *Public key*: A key that can be made public, used for computing the aggregated plaintext.

- *Private key*: Confidential key for encrypting original ciphertext.

- *Original plaintext*: The plaintext containing gradient information from a single client, which should not be accessed by other clients or servers.

- *Original ciphertext*: The ciphertext that corresponds to the original plaintext and is encrypted by a private key.

- *Aggregated plaintext*: The combined gradient information derived from multiple original ciphertexts and their corresponding public keys. In PPFL contexts, this aggregated plaintext may be shared openly among all participants.

- *Original model*: The initial HANs model distributed to clients by servers or third parties. It is potentially vulnerable to information leakage due to the absence of fully trusted distributors.

- *PPU*: A process that clients can use to transform the original model into a private model. The specific algorithm and process are detailed in Appendix E.

- *Private model*: The result of applying PPU to the original model. Each client securely stores their private model, treating it with confidentiality equivalent to private keys.

- *Public dataset*: A small, noisy dataset maintained by each client to facilitate the PPU process without exposing their private model. It serves as a proxy for PPU participation.

AHE primitives differ from traditional cryptography. We will clarify the capabilities and significance of the following attack methods in the AHE context:

- *Ciphertext-only attack (COA)*: The attacker analyzes only the ciphertext, knowing it was encrypted using AHE but without knowledge of the specific HANs model.

- *Known-model attack (KMA)*: Attacker knows the ciphertext is AHE-encrypted and has access to the original model parameters, but not the private model parameters. This corresponds to known-plaintext attacks and chosen-plaintext attacks in traditional cryptography.

- *Chosen-ciphertext attack (CCA)*: Not applicable in AHE as the encryptor cannot derive plaintext from ciphertext, making this attack infeasible in our scenario.

## 3.2 DEFINITION OF AHE

Here we further define the algorithms of AHE. It is worth noting that both the private and public keys are always real numbers rather than integers. This significantly expands key space of AHE, thereby enhancing the security of the algorithm.

1. $(pk, sk) \leftarrow \text{KeyGen}(\kappa)$. Generates a public key $pk$ and private keys $sk = \{sk_A, sk_B\}$.

2. $\vec{c} \leftarrow \text{Enc}(m, sk_A, sk_B, \psi)$. Encrypts real number $m \in [-\psi, \psi]$ using two private keys $sk_A$ and $sk_B$.

3. $m_{\text{agg}} \leftarrow \text{Agg}\left(\{\vec{c}_i\}_{i=1}^n, \{pk_i\}_{i=1}^n\right)$. Aggregates $n$ ciphertexts and outputs the sum of the plaintexts.

Each private key consists of two real numbers, $sk_A$ and $sk_B$, generated from a security parameter $\kappa$. The aggregated result only reveals the sum of the plaintexts, ensuring security as individual ciphertexts cannot be reversed.

## 3.3 USABILITY IN MODELING

In traditional HE schemes like CKKS, the error introduced by HE must be relatively small compared to the ciphertext modulus (Cheon et al., 2017). However, in the context of PPFL, our criteria can be somewhat relaxed. Our primary objective is to ensure that the difference between the homomorphically aggregate values $m_{agg}$ and the actual value $m_{real}$ does not significantly affect the model's overall performance. Specifically, we require the original model to have high performance, so that after undergoing the PPU phase, it can maintain an acceptable level of performance.

## 3.4 THREAT MODEL IN AHE SETTING

**Attack Process.** The adversary aims to exfiltrate the dataset of client $D_i$ through a three-step process:

1. **Step 1:** The adversary intercepts the encrypted messages $\vec{c}_i$ and public keys $pk_i$ transmitted between the client and the server during the PPFL process. intercept($\cdot$) is an interception algorithm capable of capturing all information transmitted through a communication channel: $(\vec{c}_i, pk_i) \leftarrow \text{intercept}(\cdot)$, where $\vec{c}_i = \text{Enc}(\theta_i, sk_{iA}, sk_{iB}, \psi)$ where $\theta_i$ represent model parameters of $client_i$

2. **Step 2:** Currently, there is no technology that can obtain plaintext information solely by analyzing the ciphertext $c$ of HANs. Consequently, an attacker attempting a COA would be unsuccessful. Instead, the attacker would likely resort to a KMA. While the attacker may have access to the Original Model, they cannot know the target's private model. To break the ciphertext, they would utilize the original model to train two models $crack_1(\cdot)$ and $crack_2(\cdot)$.

   Detailed descriptions of these architectures and information on how to obtain them will be provided in the following section. We use $\theta_{attack}$ to represent model parameters cracked by the attacker: $\theta_i^{attack1} \leftarrow crack_1(\vec{c_i}, pk_i)$ and $\theta_i^{attack2} \leftarrow crack_2(\vec{c_i})$

3. **Step 3:** Using the cracked information, the adversary attempts to reconstruct the dataset of client $D_i$. $reconstruct(\cdot)$ is an algorithm capable of reconstructing datasets based on gradients: $D_i^{attack1} \leftarrow reconstruct(\theta_i^{attack1})$ and $D_i^{attack2} \leftarrow reconstruct(\theta_i^{attack2})$

An attack is deemed successful if, upon reconstruction, either one of the two datasets by the adversary is similar to the authentic client dataset:

$$Attack\ successful. \Leftrightarrow D_i^{attack1} \simeq D_i\ \cup\ D_i^{attack2} \simeq D_i$$

We presume the adversary possesses robust reconstruction capabilities. Under these conditions, a successful attack implies the adversary's ability to effectively decrypt gradient information. It is imperative to note that we protect our private model parameters with confidentiality equivalent to that of private keys. We assume attackers cannot access these private model parameters, just as they cannot access private keys. Thus, attacks are only considered successful if the adversary achieves their objective without prior knowledge of the private model parameters.

## 3.5 PSEUDO $N$-1 COLLUSION ATTACKS

In addition to attacks and challenges targeting the model's inherent encryption capabilities, the unique characteristics of HANs may lead to two types of pseudo $N$-1 collusion attacks. These attacks attempt to overcome the limitations of traditional $N$-2 collusion attacks by leveraging additional information to achieve an effect approximating $N$-1 collusion. However, due to the PPU mechanism, their effectiveness remains significantly limited. The basic ideas behind these two attacks are:

1. **Pseudo $N$-1 Collusion Attack based on Original Model (PCAOM):** In this attack, the adversary uses another trusted client's original model to substitute for that client's private model. This is a KMA where the attacker attempts to simulate collusion among $N$-1 clients by using the publicly available original model, while in reality only $N$-2 clients are colluding.

2. **Pseudo $N$-1 Collusion Attack based on Public Dataset (PCAPD):**
   This attack utilizes the noisy public datasets information generated during the PPU process. The attacker uses this public data to approximate the behavior of another trusted client, thereby achieving an effect similar to $N$-1 collusion. This is an enhanced version of a COA.

Detailed formal definitions of these two attacks can be found in Appendix D.

## 3.6 DESIGN AND TRAINING OF HANs

Under the framework of the AHE scheme, the core of HANs lies in its carefully designed optimization objectives and training process, which work together to achieve a balance between privacy protection and accurate aggregation. This section introduces the main optimization objectives of HANs, with detailed implementation specifics available in the Appendices B and C.

The design of HANs primarily revolves around three main optimization objectives:

1. **Attacker's Optimization Objective**:

$$O_{Enc} = argmax_{\theta_{client}}((L_{Eve}^{client}(\theta_{client}, \theta_{Eve}^{client})) + \hat{L}_{Eve}^{client}(\theta_{client}, \hat{\theta}_{Eve}^{client}))$$

   This objective aims to maximize the attacker's error in reconstructing the original data. A larger value of the loss indicates stronger privacy protection.

2. **Aggregation Optimization Objective**:

$$O_{agg} = argmin(L_{agg}(\theta_{Alice}, \theta_{Bob}, \theta_{Carol}, \theta_{Agg}))$$

This objective ensures that the encrypted data can still be accurately aggregated. A smaller value of the loss indicates higher aggregation accuracy.

3. **Comprehensive Optimization Objective**:

$$O_{Enc} = argmin_\theta(\lambda \mathbb{E}_\theta + \sum_{i \in Clients} (max(0, \gamma - L_{Eve}^i(\theta_i, \theta_{Eve}^i))$$
$$+ max(0, \gamma - \hat{L}_{Eve}^i(\theta_i, \hat{\theta}_{Eve}^i))))$$

This objective balances privacy protection and aggregation accuracy. Here, $\gamma$ represents a privacy coefficient controlling the trade-off between security and accuracy, and $\lambda$ is a balancing parameter for aggregation accuracy.

The optimization objectives in HANs are designed to balance privacy protection and model usability throughout the training process. By employing a multi-stage optimization strategy—encompassing computational pre-training, security enhancement, security assessment, and performance-security balancing, HANs ensures robust security without compromising performance. Detailed descriptions of the optimization objectives, the multi-stage optimization process can be found in Appendix B an C.

## 3.7 PPU

To further enhance privacy protection, we have designed a PPU process, which includes two stages:

- **CPPU:** The main goal of the CPPU stage is to reduce the risk of model exposure, especially in scenarios involving multi-party collaboration. Each client generates its own private data and collects the latest public datasets from other clients. By combining private data with samples from other clients, the client creates a training set to update the model. To minimize the leakage of private model information, we gradually add noise to the public dataset, with the noise intensity increasing over time. This noise not only prevents potential future inference attacks but also protects the privacy of other clients.

- **IPPU:** The IPPU stage further enhances security by performing multiple rounds of independent updates using only the client's private data, thereby weakening the correlation between the public dataset and the client's encryption model. This process significantly reduces the possibility for attackers to infer sensitive information about the encryption model, even if they possess advanced techniques to analyze changes in the public dataset.

The PPU process balances privacy protection and model performance. Although the update process may lead to a slight decrease in model performance, the combination of CPPU and IPPU effectively enhances the overall security of the system.

For a detailed implementation of the PPU steps and algorithms, please refer to the Appendix E.

## 3.8 SECURITY DISCUSSION OF HANs

Neural network-based cryptosystems, such as HANs, preclude conventional mathematical security proofs due to their inherent opacity. Nonetheless, we conduct an indirect security assessment by examining the constraints on attacker-accessible information.

Appendix G elucidates the potential information exposure across HANs' lifecycle phases: training, PPU, and operational deployment. Our analysis indicates that HANs' architectural design substantially mitigates the efficacy of exploitable information (e.g., model parameters, public datasets, cryptographic keys, and ciphertexts) in practical attack scenarios.

The observed limitations on actionable information, in conjunction with HANs' empirically demonstrated resilience against diverse attack vectors, provide compelling indirect evidence for its security robustness and operational reliability.

Table 2: Performance and attack resistance of HANs

| | HANs | Atk 1 | Atk 1 (Dbl) | Atk 2 | Atk 2 (Dbl) |
|---|---|---|---|---|---|
| Average | 0.000009 | 0.0644 | 0.0653 | 0.0041 | 0.0013 |
| Maximum differences | 0.001772 | 1.4250 | 1.6599 | 0.1937 | 0.1179 |
| After CPPU | | | | | |
| Average | 0.0002 | 0.1037 | 0.1048 | 0.0691 | 0.0633 |
| Maximum differences | 0.0412 | 1.4341 | 1.2353 | 1.5290 | 1.7215 |
| After IPPU | | | | | |
| Average | 0.0004 | 0.1025 | 0.1055 | 0.1060 | 0.1047 |
| Maximum differences | 0.0569 | 1.5140 | 1.3213 | 1.4687 | 1.9999 |

Table 3: Aggregation differences and their impact on FL accuracy using the HANs Model

| | MNIST | FashionMNIST | CIFAR-10 |
|---|---|---|---|
| Accuracy difference | +0.48% | -0.27% | -1.35% |
| Average difference | 0.0047 | 0.0056 | 0.0097 |
| Standard deviation of average difference | 0.0003 | 0.0006 | 0.0016 |
| Maximum differences | 0.1219 | 0.1904 | 0.2941 |
| Standard deviation of Maximum differences | 0.0367 | 0.0461 | 0.0623 |

# 4 EXPERIMENTAL ANALYSIS

This section aims to validate the accuracy, security, and efficiency of HANs. All experiments were conducted using an A800 GPU. We have included the detailed design of the loss function, along with the specific training, distribution processes and PPU, in the Appendix B, C, and E.

To improve experimental efficiency, we ensured that the encryption model structure used by each client was consistent. However, the attacker models were allowed to vary in structure to accommodate different attack strategies.

The encryption model consists of linear layers, convolutional layers, and residual blocks. The initial linear layer expands the dimensionality of the plaintext and private keys, while the convolutional layers obscure the relationship between them. Multiple residual blocks further enhance the complexity of the input transformation, and the output layer compresses the data to the target ciphertext length. The attacker models mirror the architecture of the encryption model, with input dimensions adjusted to accommodate the ciphertext input.

We employed the AdamW optimizer with a learning rate of 1e-5 and weight decay of 1e-6, combined with a cosine annealing scheduler to ensure training stability and generalization. Training data were generated by simple addition, enabling the model to handle diverse inputs and avoid overfitting.

## 4.1 TRAINING OPTIMIZATION AND PPU ENHANCEMENTS

We evaluated four attacker types: Atk 1 (using ciphertext and public keys), Atk 2 (using only ciphertext), and their double residual block versions, Atk 1 (Dbl) and Atk 2 (Dbl), as shown in Table 2.

For the encryption model, lower **Average** and **Maximum Differences** indicate better performance. For attackers, higher **Average** signify greater difficulty in data reconstruction.

The results show that Atk 1 faces greater challenges in data reconstruction compared to Atk 2, likely due to the added complexity from public key information. Even with increased complexity in Atk 1 (Dbl) and Atk 2 (Dbl), their performance improvements were marginal, suggesting that simply increasing model complexity is insufficient to break HANs' encryption mechanism.

The PPU process further enhanced security. Both CPPU and IPPU stages progressively increased the difficulty for attacker models, as evidenced by higher average and maximum differences. The narrowing performance gap between standard and double versions of the attacks further underscores the limitations of relying solely on increased model complexity to breach HANs' security.

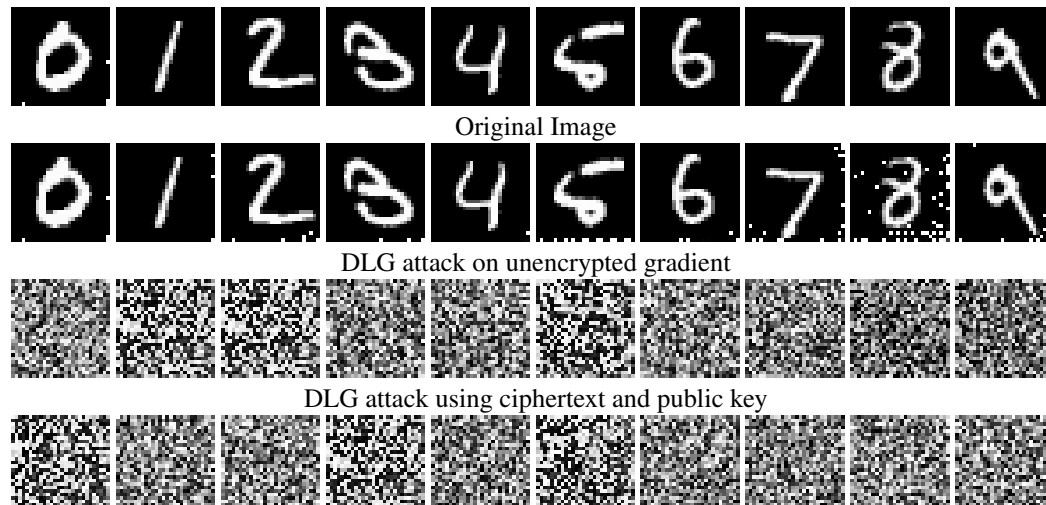

Original Image

DLG attack on unencrypted gradient

DLG attack using ciphertext and public key

DLG attack using ciphertext

The figure illustrates: (1) original dataset images; (2) reconstructions from DLG attacks on unencrypted gradients; (3-4) results of DLG attacks on HANs-encrypted gradients, using ciphertext with public keys and ciphertext alone, respectively.

Figure 1: DLG Attack

Table 4: PCAOM and PCAPD comparison (2,000 samples)

| | PCAOM MAD | PCAOM Var | PCAPD MAD | PCAPD Var |
|---|---|---|---|---|
| | 0.31067 | 0.14935 | 0.30340 | 0.14535 |
| Attack Examples | | | | |
| Orig. Val. | PCAOM Est. | PCAOM Diff. | PCAPD Est. | PCAPD Diff. |
| 0.29563 | 0.51904 | 0.22340 | 0.51725 | 0.22162 |
| 0.04339 | 0.19841 | 0.15501 | 0.20070 | 0.15730 |
| -0.08219 | 0.11783 | 0.20002 | 0.12819 | 0.21038 |
| 0.00916 | 0.42505 | 0.41589 | -0.14483 | 0.15399 |
| -0.00047 | 0.17528 | 0.17575 | 0.18152 | 0.18199 |

**Note:** MAD = Mean Absolute Difference, Var = Variance, Est. = Estimated Value, Diff. = Difference

Overall, these results demonstrate the stability and attack resistance of the encryption model across different scenarios, showing that it effectively resists attempts to enhance attack success through increased computational complexity. While these metrics offer valuable insights into model performance and security, they do not provide absolute thresholds for meeting System Goals A.2. Therefore, further investigation in practical FL scenarios is required to fully evaluate the performance and security of HANs.

## 4.2 PERFORMANCE AND SECURITY ANALYSIS OF HANS IN FL

Table 3 presents a comparison between traditional additive aggregation and HANs aggregation on the MNIST (Deng, 2012), FashionMNIST (Xiao et al., 2017), and CIFAR-10 (Krizhevsky et al., 2009) datasets.

The **Accuracy difference** shows the impact of HANs on model performance. On MNIST, there is a 0.48% accuracy improvement, which could be due to the additional noise introduced during aggregation acting as a form of regularization on simpler datasets. However, there is a slight drop in accuracy for FashionMNIST (-0.27%) and CIFAR-10 (-1.35%).

The **Average difference** and **Maximum differences** quantify the discrepancies between parameters aggregated using HANs and traditional methods. Despite larger differences in some parameters,

overall model performance remains nearly unaffected, demonstrating that HANs can maintain strong model performance while ensuring privacy.

To validate the security of our proposed scheme, we employ simple models in conjunction with the original DLG attack. While recent research has advanced to more complex models and efficient reconstruction techniques (Zhao et al., 2020; Geiping et al., 2020), the use of DLG on simpler models is sufficient for our security verification purposes.

We evaluated HANs' defense against DLG attacks using the MNIST dataset, which is known to be vulnerable (Zhu et al., 2019). Without HANs, DLG attacks were effective, but with HANs encryption, dataset reconstruction was unsuccessful (see Figure 1).

### 4.3 Resistance to Pseudo $N$-1 Collusion Attacks

In evaluating the security of HANs, we conducted experiments on pseudo $N$-1 collusion attacks. Table 4 presents the results of the PCAOM and PCAPD.

These attack methods attempt to simulate the effect of $N$-1 collusion attacks, but their effectiveness is significantly limited due to the PPU mechanism. The experimental results show that after implementing PPU, PCAOM has a MAD of 0.31067, while PCAPD has a MAD of 0.30340. Notably, before the implementation of PPU, the result of PCAOM was equivalent to the Average value of HANs, indicating that the PPU mechanism effectively enhanced the system's security.

The lower half of the table 4 provides attack results for five specific samples from different orders of magnitude. Notable differences between the estimated and original values can be observed, ranging from 0.15399 to 0.41589. This further confirms the effectiveness of HANs in resisting these advanced attacks.

The similar performance of both attack methods suggests that the PPU mechanism successfully limits the amount of potentially leaked information, thereby enhancing the overall security of the system.

Table 5: Performance metrics of HANs for various batch sizes (results based on 1000 experiments)

| Batch Size | Batch Encryption Time | Batch Aggregation Time | Key Generation Time |
|---|---|---|---|
| 100,000 | 0.019554s (±0.001629) | 0.017444s (±0.000108) | 0.000028s (±0.000008) |
| 200,000 | 0.035329s (±0.000027) | 0.035380s (±0.000013) | 0.000029s (±0.000008) |
| 300,000 | 0.053281s (±0.003922) | 0.053462s (±0.004456) | 0.000031s (±0.000012) |

### 4.4 Operating Efficiency

To evaluate the computational efficiency of HANs, we conducted a series of experiments assessing encryption time, aggregation time, and communication overhead across various scenarios. Table 5 shows the performance metrics of HANs for different batch sizes.

The encryption time for a batch size of 100,000 (0.019554s) is less than 2 times that of 200,000 (0.035329s), indicating that for smaller batch sizes, the GPU's computational capacity is not fully utilized. Therefore, we use the encryption and aggregation times for the batch size of 300,000 to extrapolate the performance for 3,000 ciphertexts.

Table 6 summarizes our findings, juxtaposing HANs against the SecFed scheme (Cai et al., 2023).

Our results demonstrate that HANs significantly outperforms SecFed in terms of computational efficiency. For a batch of 3,000 ciphertexts, HANs completes encryption and aggregation in just 0.00107 seconds, compared to SecFed's 6.5 seconds. This represents a remarkable 6,075-fold speedup, highlighting the exceptional computational efficiency of our approach. The significant improvement in computational efficiency comes at a cost, specifically a 29.2-fold increase in communication overhead compared to SecFed.

The dramatic performance improvement can be attributed to HANs' ability to leverage GPU parallel computing capabilities, a benefit inherent to its neural network-based architecture. This allows

Table 6: Performance comparison between HANs and SecFed

| Metric | HANs | SecFed |
|---|---|---|
| **Encryption and Aggregation Performance (3,000 Ciphertexts)** | | |
| Total Time | 0.00107s | 6.5s |
| Speedup | 6,075x | 1x (baseline) |
| **Communication Overhead (One-Round)** | | |
| Model Size: 616,420 | 232.8 MB | 2.6 MB |
| Model Size: 7,027,860 | 884.7 MB | 30.2 MB |
| Overhead Increase | 29.2x | 1x (baseline) |

HANs to efficiently process large volumes of parameters simultaneously, resulting in significantly reduced computation time.

While these results are promising, it is essential to acknowledge the inherent limitations in our comparative analysis. As the pioneering approach using neural networks for MK-HE, our comparison with traditional methods encounters certain constraints. The reported 6.5-second runtime for SecFed may underestimate its operational complexity, as additional procedures, such as multiple refresh operations, could potentially extend its execution time, potentially amplifying HANs' efficiency gains. Conversely, the lack of information about SecFed's GPU acceleration capabilities, and the potential challenges in adapting their scheme to GPU architectures, prevents us from replicating their results in an equivalent computing environment, introducing some uncertainty into our comparative findings.

## 5 FUTURE WORK

This work introduces a novel framework for privacy-preserving federated learning by utilizing neural networks to emulate multi-key homomorphic encryption. While the results demonstrate the feasibility and foundational performance of the proposed approach, several valuable directions remain for further exploration. One promising area is optimizing communication overhead, including reducing unnecessary ciphertext transmissions and improving the efficiency of aggregation mechanisms, to enhance practical applicability. Expanding evaluations to include more diverse datasets, such as text or multi-modal data, and more complex model architectures, such as transformers or large-scale neural networks, will help establish the method's scalability and robustness. Additionally, developing rigorous privacy analyses, including systematic methodologies and formal security proofs, will provide a stronger theoretical foundation and more comprehensive insights into the trade-offs between privacy guarantees and model performance. These directions represent critical steps toward advancing the applicability and impact of neural network-based privacy-preserving federated learning.

## 6 CONCLUSION

This work introduces Homomorphic Adversarial Networks (HANs) with Aggregatable Hybrid Encryption for Privacy-Preserving Federated Learning (PPFL). HANs leverage neural networks to emulate multi-key homomorphic encryption, offering a novel approach that balances privacy, performance, and efficiency. Our method enables independent key generation and aggregation without collaborative decryption, while resisting $N$-2 client collusion. The innovative Privacy-Preserving Update mechanism enhances security through private model updates, effectively mitigating potential vulnerabilities in the initial public model. EExperimental results demonstrate HANs' ability to maintain model accuracy within 1.35% of non-private federated learning. HANs also significantly outperform traditional multi-key homomorphic encryption schemes, achieving a 6,075-fold increase in computational efficiency. The introduction of these neural network-based protocols not only improves the practical implementation of PPFL but also opens new research directions in federated learning privacy protocols and neural network-based cryptography.

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

## A  PROBLEM OVERVIEW

### A.1  PROBLEM SETTING

In the setting of FL, there are two primary roles: (1) $Clients$, who possess local training datasets and are responsible for completing local model training. $Clients$ have the obligation to ensure the privacy and security of the dataset. (2) The $Server$, responsible for coordinating with $Clients$ to update global model parameters, also initializes the model and global hyperparameter settings.

Suppose that we have $m$ separate $Clients$. Each $Client$ is represented by $C_i$, where $i \in [1, m]$ and $Client$ $C_i$ has a local training dataset $\mathcal{D}_i$. In each step, there are three sub-steps:

1. **Broadcast.** $Server$ broadcasts the current global model parameters $w^{t-1}$ to each $Client$ $C_i$, where $t$ represents the index of the current iteration round.

2. **Local training.** Each $Client$ $C_i$ receives global model parameters $w^{t-1}$ and using local training datasets $\mathcal{D}_i$ to obtain the new local model parameters $w_i^t$ in parallel and sends the local model parameter $w_i^t$ back to the $Server$. During the updating step, the $Client$ typically employs stochastic gradient descent for local epochs. In scenarios where communication cost is not a primary concern, setting local epochs to 1 can be an effective approach (McMahan et al., 2017).

3. **Aggregation.** The $Server$ receives all model parameters $\{w_1^t, w_2^t, \ldots, w_m^t\}$ from the $Client$ and aggregates them into global parameters $w^t$ by averaging them.

**Definition of $\delta$-accuracy loss.** Suppose that $\bar{\mathcal{M}}$ is a deep learning model training on datasets $\mathcal{D}$, where $\mathcal{D} = \mathcal{D}_i \cup \mathcal{D}_2 \cup \cdots \cup \mathcal{D}_m$. We use $\bar{f}$ to denote the accuracy of model $\bar{\mathcal{M}}$. For FL, $\hat{\mathcal{M}}$ denotes the model after all train rounds, and its corresponding accuracy is $\hat{f}$. We say that it is $\delta$-accuracy loss, if it satisfies $\bar{f} - \hat{f} < \delta$.

### A.2  SYSTEM GOALS

1. **Input privacy.** Our objective is to preserve the privacy of the client dataset $\mathcal{D}_i$ during all processes, even if attackers gain access to either partial true gradient information or noisy gradient information $w^{attack}$, which is insufficient to reconstruct $\mathcal{D}_i$.

2. **Model utility.** After several rounds of encryption and aggregation, the final global parameters of the model $w^{final}$ are accurately computed, ensuring that the model can be used as intended.

We assume that all parties involved in the agreement will correctly complete model training and aggregation according to the FL agreement.

### A.3  THREAT MODEL

We consider the threat model where the adversary aims to steal the gradient information $w^t$ and $w_i^t$ transmitted between the client and the server, and then use it to reconstruct the dataset $\mathcal{D}_{attack}$ to match a specific client's dataset $\mathcal{D}_i$.

## B  HANs TRAINING DESIGN: LOSS FUNCTION FORMULATION AND RATIONALE

In this section, we provide a complete and rigorous derivation of the optimization objectives for HANs, expanding upon the high-level summary presented in Section 3.6. The derivation includes the mathematical formulation of the encryption and aggregation processes, as well as the multi-stage optimization strategy employed to balance privacy and performance.

Firstly, we would like to clarify that all distances mentioned in this paper refer to the Manhattan distance. During model training, we employ MSEloss $L_m$ as our loss function, which enhances our ability to train the model effectively. However, when evaluating the model, we utilize L1loss, as it allows for a more intuitive analysis of errors.

We have formally defined the goals of each party in the previous section, and now we will present the specific training methods. We will use $\theta$ representing model parameters in HANs.

The adversary Eve's goal is simple: to accurately reconstruct $w$ to achieve the attacker's objective. We will employ two attack methodologies, namely attacks with and without the use of a public key. The utilization of dual attack methods can ensure the resilience of the attacker. Additionally, if we can effectively thwart both attacks simultaneously, it will demonstrate the defensive efficacy of our solution.

To ensure the security of each $Client$'s data, where $Clients = \{Alice, Bob, Carol\}$, we have designed separate attacker for each $Client$. We train each attacker independently. We use the Alice's attacker as the example for discussion. We denote the attacker attacking Alice's output on input C without the public key as $Attack(\theta_{Eve}^{Alice}, C)$, and the attacker attacking Alice's output with the public key as $Attack(\theta_{Eve}^{Alice}, (sk_1 + sk_2), C)$. The loss function for the two attack models is designed as follows:

$$L_{Eve}^{Alice}(\theta_{Alice}, \theta_{Eve}^{Alice}, W_{Alice}, sk_{1A}, sk_{2A}) = L_m(W, Attack(\theta_{Eve}^{Alice}, Enc_{sk_{1A}, sk_{2A}}(W_{Alice})))$$

$$\hat{L}_{Eve}^{Alice}(\theta_{Alice}, \hat{\theta}_{Eve}^{Alice}, W, sk_{1A}, sk_{2A}) = L_m(W, Attack(\hat{\theta}_{Eve}^{Alice}, \qquad\qquad (1)$$
$$(sk_{1A} + sk_{2A}), Enc_{sk_{1A}, sk_{2A}}(W_{Alice})))$$

The sole distinction between $L_{Eve}^{Alice}$ and $\hat{L}_{Eve}^{Alice}$ lies in the fact that the model $\hat{\theta}_{Eve}^{Alice}$ employed in $\hat{L}_{Eve}^{Alice}$ incorporates a public key $pk_A = (sk_{1A} + sk_{2A})$ as an additional input parameter. Intuitively, the loss function signifies the degree to which Eve is incorrect in terms of the model parameter $w$. Given that the model parameters and public/private key pairs for encryption are randomly generated during the training process, the aforementioned loss function can be interpreted as the expected value distribution on parameters and private key pairs.

$$L_{Eve}^{Alice}(\theta_{Alice}, \theta_{Eve}^{Alice}) = \mathbb{E}_{(W, sk_{1A}, sk_{2A})}(L_m(W, Attack(\theta_{Eve}^{Alice}, Enc_{sk_{1A}, sk_{2A}}(W_{Alice}))))$$

$$\hat{L}_{Eve}^{Alice}(\theta_{Alice}, \hat{\theta}_{Eve}^{Alice}) = \mathbb{E}_{(W, sk_{1A}, sk_{2A})}(L_m(W, Attack(\hat{\theta}_{Eve}^{Alice}, \qquad\qquad (2)$$
$$(sk_{1A} + sk_{2A}), Enc_{sk_{1A}, sk_{2A}}(W_{Alice}))))$$

For attackers, they aim to derive the 'optimal attacker model' by minimizing the loss for each $Client$ included in the $Clients$ list.

$$O_{Eve}(\theta_{Eve}^{client}) = argmin_{\theta_{Eve}^{client}}(L_{Eve}^{client}(\theta_{Alice}, \theta_{Eve}^{client}))$$

$$O_{Eve}(\hat{\theta}_{Eve}^{client}) = argmin_{\hat{\theta}_{Eve}^{client}}(\hat{L}_{Eve}^{client}(\theta_{Alice}, \hat{\theta}_{Eve}^{client}))$$

The preceding discussion focuses on optimization methods for attackers. As for encryption, intuitively, it can be viewed as a hindrance to the attackers' goals. These goals can be achieved by maximizing the loss for each $client$ in the $Clients$ list, thus realizing the optimal encryption model:

$$O_{Enc}(\theta_{Alice}) = argmax_{\theta_{client}}((L_{Eve}^{client}(\theta_{client}, \theta_{Eve}^{client})) + \hat{L}_{Eve}^{client}(\theta_{client}, \hat{\theta}_{Eve}^{client}))$$

Via the design previously detailed, we have successfully fulfilled the need for privacy protection. Beyond reaching the defense efficacy which satisfies the objective of privacy, the encryption model also needs to comply with the aggregation accuracy standards to accomplish the error goal. Consequently, it is crucial to delineate the following loss function to adhere to the aggregation prerequisites:

$$L_{agg} = L_m((W_{Alice} + W_{Bob} + W_{Carol}), Agg_{pk_A, pk_B, pk_C}(Enc_{sk_1, sk_2}(W)))$$

The foregoing presents a succinct delineation, while a more comprehensive depiction for each symbol is as follows.

$$L_{agg} = L_{agg}(\theta_{Alice}, \theta_{Bob}, \theta_{Carol}, \theta_{Dec}, W_{Alice}, W_{Bob}, W_{Carol}, \qquad\qquad (3)$$
$$sk_{1A}, sk_{2A}, sk_{1B}, sk_{2B}, sk_{1C}, sk_{2C})$$

$$Enc_{sk_1,sk_2}(W) = (Enc_{sk_{1A},sk_{2A}}(W_{Alice}), Enc_{sk_{1B},sk_{2B}}(W_{Bob}), Enc_{sk_{1C},sk_{2C}}(W_{Carol}))$$

$$pk_i = sk_{1i} + sk_{2i}$$

This component is also perceived as an expectation that is predicated on the distribution during our actual training process.

$$L_{agg}(\theta_{Alice}, \theta_{Bob}, \theta_{Carol}, \theta_{Agg}) = \mathbb{E}_\theta = \mathbb{E}_{W,sk_1,sk_2}(L_m((W), Agg_{pk_A,pk_B,pk_C}(Enc_{sk_1,sk_2}(W))))$$

Similar to the approach of attackers, the optimal model can be realized by minimizing losses to achieve precision.

$$O_{agg} = argmin(L_{agg}(\theta_{Alice}, \theta_{Bob}, \theta_{Carol}, \theta_{Agg}))$$

Typically, the HANs could be achieved as follows:

$$O_{Enc} = argmin_\theta(\mathbb{E}_\theta - (\sum_{i \in Clients} (L^i_{Eve}(\theta_i, \theta^i_{Eve}) + (\hat{L}^i_{Eve}(\theta_i, \hat{\theta}^i_{Eve}))))) \quad (4)$$

Nevertheless, the practical application might encounter specific difficulties. For example, amplifying the latter part could result in a never-ending quandary. This situation might cause an excessive focus on defense capabilities and disregard for precision considerations during the model training phase. As mentioned earlier, we do not necessitate such robust defensive abilities. Therefore, we utilize the subsequent strategies to train the model.

$$O_{Enc} = argmin_\theta(\lambda\mathbb{E}_\theta + \sum_{i \in Clients} (max(0, \gamma - L^i_{Eve}(\theta_i, \theta^i_{Eve})) + max(0, \gamma - \hat{L}^i_{Eve}(\theta_i, \hat{\theta}^i_{Eve}))))$$

$$(5)$$

This objective balances privacy protection and aggregation accuracy. Here, $\gamma$ represents a security coefficient used during the training process to guide the model towards a desired level of privacy protection. It's important to note that $\gamma$ is not a strict requirement for the final system performance, but rather a training parameter to help achieve a balance between security and utility.

## C  HANs Training Implementation: A Multi-stage Optimization Process

In the previous chapter, we elaborated on the design of the HANs model, including the formulation of the loss function and the rationale behind its design. These design considerations laid the theoretical foundation for our training process. However, in practical implementation, we discovered that directly applying the designed optimization objective 5 for training might lead to two main challenges:

- **Imbalance between Security and Usability**: The model might overemphasize Input Privacy while neglecting Usability in Modeling, resulting in the $Enc$ in HANs generating ciphertexts unrelated to the plaintext. In this case, while security is ensured, the model's practical utility is severely compromised.

- **Insufficient Aggregation Functionality**: Even when $Enc$ generates ciphertexts that contain plaintext meaning and are sufficiently secure, the $Aggregate$ model might not be adequately trained to complete the aggregation task. This leads to the entire system being unable to effectively process and integrate encrypted data from multiple sources.

To address these challenges, we have decomposed the training process into five crucial stages:

1. **Computational Pre-training**: Utilizing optimization formula 4, aiming to satisfy Usability in Modeling.

2. **Security Enhancement Training**: Employing optimization formula 5, with the objective of achieving Input Privacy while maintaining Usability in Modeling.

3. **Security Assessment**: This phase fixes all $Enc$ models and the $Aggregate$ in HANs, continuing to train each Attack model until convergence. As we have not yet determined a definitive security boundary, it is necessary to simulate it in an FL scenario for additional security confirmation.

4. **Performance-Security Balance Adjustment**: HANs models trained through the first two stages often prioritize security at the expense of performance. Therefore, we conduct small-scale training using loss function A, followed by a final security validation on the trained HANs model. If it fails, we repeat the performance-security balance adjustment; if it passes, we proceed to the fifth stage.

5. **Aggregation Alignment**: Having ensured the security of individual $Enc$ models through previous training, this stage fixes all $Enc$ models and trains the $Aggregate$ model using optimization formula 4 until convergence, concluding the comprehensive training process for the HANs model.

# D  PSEUDO $N$-1 COLLUSION ATTACKS

PPFL scenarios based on multi-key homomorphic encryption typically can only resist $N$-2 collusion attacks. This is because if only one client remains honest and trustworthy, colluding clients can easily obtain that client's real data by subtracting their uploaded data from the aggregated model gradients. This scenario is not one that multi-key homomorphic encryption is designed to defend against, and our method is no exception. However, due to the unique characteristics of HANs, attackers may potentially employ two types of pseudo $N$-1 collusion attacks. We will now formally define these two attacks.

## D.1  PSEUDO $N$-1 COLLUSION ATTACK BASED ON THE ORIGINAL MODEL (PCAOM)

PCAOM is a KMA. Let $clients = client_1, \ldots, client_N$ be a set of $N$ $clients$ in a FL system using HANs. This attack can be described in the following steps:

1. **Initial Setup:**
   - A trusted client Alice ($client_A \in clients$) encrypts a gradient message $m_A$: $c_A = Enc(m_A, sk_{A1}, sk_{A2})$
   - Alice's public key $pk_A = sk_{A1} + sk_{A2}$ is transmitted and intercepted.

2. **Attacker's Preparation:**
   - The $N$-2 colluding attackers acquire the latest aggregation model $Aggregation()$.
   - The attacker ($client_{att} \in clients \setminus \{client_A, client_B\}$) generates and encrypts $m_{att}$:

   $$c_{att} = Enc(m_{att}, sk_{att1}, sk_{att2})$$

   - The attacker's public key: $pk_{att} = sk_{att1} + sk_{att2}$

3. **Exploitation of Bob's Original Model:**
   - The attacker uses Bob's ($client_B \in clients$) original model to encrypt $m_B$:

   $$c_B^{origin} = Enc_{origin}(m_B, sk_{B1}, sk_{B2})$$

   - The corresponding public key: $pk_B^{origin} = sk_{B1} + sk_{B2}$

4. **Aggregation:**

   $$m_{agg} \leftarrow Aggregate(\vec{c_A}, \vec{c_B^{origin}}, \vec{c_{att}}, pk_A, pk_B^{origin}, pk_{att})$$

5. **Guessing Alice's Message:**

   $$m_A^{guess} = m_{agg} - m_B - m_{att}$$

The goal of PCAOM is to approximate $m_A^{guess} \approx m_A$.

### D.2 Pseudo $N$-1 Collusion Attack based on Public Dataset (PCAPD)

This represents an enhanced version of a COA, where the attacker has knowledge of the decryption model and access to noisy plaintexts corresponding to known ciphertexts. Let $clients = client_1, \ldots, client_N$ be a set of $N$ clients in a FL system using HANs. The PCAPD proceeds as follows:

1. **Initial Setup:**
   - A trusted client Alice ($client_A \in clients$) encrypts a message $m_A$: $c_A = Enc(m_A, sk_{A1}, sk_{A2})$
   - Alice's public key $pk_A = sk_{A1} + sk_{A2}$ is transmitted and intercepted.

2. **Attacker's Preparation:**
   - The $N$-2 colluding attackers acquire the latest aggregation model.
   - An attacker ($client_{att} \in clients \setminus \{client_A, client_B\}$) generates and encrypts $m_{att}$:
   $$c_{att} = Enc(m_{att}, sk_{att1}, sk_{att2})$$
   - The attacker's public key: $pk_{att} = sk_{att1} + sk_{att2}$

3. **Exploitation of Bob's Public Dataset:**
   - The attacker obtains Bob's ($client_B \in clients$) public dataset from the last round of the PPU process.
   - From this dataset, the attacker extracts:
     - A noisy plaintext $m_B^{pub} = m_B + noise$, where $noise$ is unknown to the attacker
     - The corresponding ciphertext $c_B^{pub} = Enc(m_B, sk_{B1}, sk_{B2})$
     - The public key $pk_B^{pub} = sk_{B1} + sk_{B2}$

4. **Aggregation:**
   $$m_{agg} \leftarrow Aggregate(\vec{c_A}, \vec{c_B^{pub}}, \vec{c_{attack}}, pk_A, pk_B^{pub}, pk_{attack})$$

5. **Guessing Alice's Message:**
   $$m_A^{guess} = m_{agg} - m_B^{pub} - m_{attack}$$

   Note that this guess includes an error term due to the noise in $m_B^{pub}$.

The goal of PCAPD is to approximate $m_A^{guess} \approx m_A$, exploiting the public dataset information from the PPU process, including the noisy plaintexts and their corresponding ciphertexts.

## E   PPU Mechanism

After completing the training of HANs or opting to use a publicly trained HANs model, we distribute the models to client endpoints. At this stage, the encryption models employed by each client are public rather than private. To address this vulnerability, we implement a PPU mechanism for the encryption models. The PPU process consists of two phases:

### E.1   CPPU

The CPPU phase, as outlined in Algorithm 2, primarily aims to mitigate PCAOM. The process involves:

- Each client generates its own private data and collects the latest public datasets from other clients.
- The client creates a training set by combining its private dataset with samples from the public datasets.
- To minimize the exposure of private model information, the public datasets are kept minimal, using a sample-with-replacement algorithm (Algorithm 1) to create the training set.

A crucial aspect of CPPU is the addition of sufficient noise to the public dataset. This noise, which increases in intensity over time, should be greater than Gaussian noise $G(0, 10^{-2})$ (Zhu et al., 2019), serves a dual purpose:

- Protects against potential future techniques that might infer model parameters from public dataset changes.
- Safeguards other client endpoints.

We require trusted clients to honestly add this noise, as it is essential for the overall security of the system. It's worth noting that for malicious clients, the addition or omission of noise does not affect their attack capabilities.

### E.2 IPPU

The IPPU phase, as detailed in Algorithm 3, further enhances security by:

- Preventing PCAOM attacks.
- Diminishing the correlation between the public dataset and the $client$'s encryption model.

This process involves multiple rounds of independent updates for each client, using only their private data and the final public datasets from the CPPU phase. By doing so, IPPU significantly reduces the potential for adversaries to infer sensitive information about the encryption model, even if they possess advanced techniques for analyzing public dataset changes.

### E.3 TRADE-OFF AND SYSTEM CONSISTENCY

It is important to acknowledge that the PPU mechanism is a trade-off between model performance and security. By implementing these updates, clients sacrifice some model performance to gain enhanced security.

## F DETAILED EXPLANATION OF DLG ATTACK

In this section, we provide a detailed recapitulation of the DLG attack as originally proposed. Subsequently, Algorithm 4 and 5 elucidate how an adversary can deploy this attack within our specific scenario.

The DLG (Zhu et al., 2019) is a privacy attack method targeting distributed machine learning systems. This method can reconstruct the original training data using only the shared gradient information.

The core idea of the DLG attack is to optimize "dummy" inputs and labels to produce gradients that are as close as possible to the target real gradients. When the optimization converges, this "dummy" data becomes very close to the original training data:

$$\textit{Attack successful.} \Leftrightarrow x' \simeq x \ \cap \ y' \simeq y$$

Specifically, given a machine learning model $F(x; W)$, where $x$ is the input data and $W$ is the model parameters, and assuming we know the gradient $\nabla W$ from a certain training iteration, the goal of the DLG attack is to find a pair of input $x'$ and label $y'$ such that:

$$\arg \min_{x', y'} ||\nabla W' - \nabla W||^2$$

where $\nabla W' = \frac{\partial L(F(x', W), y')}{\partial W}$ is the gradient produced by $x'$ and $y'$.

## G SECURITY DISCUSSION OF HANs

In this section, we will delve into an in-depth discussion of the model's security. First, we further clarify the attacker's objectives against HANs. The attacker's goal for AHE is shown in 1. To

achieve this goal, the attacker aims to minimize the guessing difference to ensure the effectiveness of data reconstruction attacks. The guessing difference is defined as:

$$\text{Guessing Difference} = |m_{\text{guess}} - m_{\text{real}}| \tag{6}$$

where $m_{\text{guess}}$ represents the attacker's guessed value, and $m_{\text{real}}$ represents the actual value.

Currently, we have not determined a specific security threshold for this difference. According to Zhu et al. (2019), Gaussian noise greater than $10^{-2}$ may significantly affect the success rate of reconstruction attacks. However, this conclusion is based solely on Gaussian distributions and does not fully consider the potential impacts of other probability distributions.

To achieve the attack objective, attackers will employ various strategies to implement attacks. Although we cannot enumerate all possible attack methods, we can discuss the difficulty of attacks and the security of HANs by analyzing the information potentially exposed to attackers. Assuming HANs is used by at least two honest $clients$, $client_A$ and $client_B$, we will discuss the scenario where an attacker attempts to obtain information from $client_A$.

### G.1 INFORMATION ACCESSIBLE TO ATTACKERS

Throughout the lifecycle of HANs, attackers may gain access to the following information:

1. Training phase: $\{\theta_{\text{Enc}}, \theta_{\text{Agg}}, \theta_{\text{Attack}_1}, \theta_{\text{Attack}_2}\}$
   These parameters represent the original encryption model, original aggregation model, and attack model parameters.

2. PPU process:
   - Noisy public datasets $\{D_{\text{pub}}^i\}_{i=1}^N$
   - Aggregation model parameters after each update $\{\theta_{\text{Agg}}^t\}_{t=1}^T$
   - Gradients derivable from the original model for each update $\{\nabla\theta_{\text{Agg}}^t\}_{t=1}^T$

3. HANs usage phase: $\{pk_i, c_i\}_{i=1}^N, m_{\text{agg}}$ where $pk_i$ and $c_i$ represent the public key and ciphertext of the $i$-th $client$, respectively, and $m_{\text{agg}}$ represents the aggregated value of the ciphertexts.

It is worth noting that although attackers may obtain the above information, they cannot directly access the plaintext $m_A$ of $client_A$'s private dataset unless one of the following conditions is met:

1. Obtain the noise-free plaintext $m$ corresponding to ciphertexts of $client_B$ in the public dataset used by $client_A$, thereby implementing an attack similar to PCAPD.

2. Obtain at least two same noise-free plaintexts $m_A^1 = m_A^2$ corresponding to ciphertexts $c_A^1, c_A^2$ in the noisy dataset published by $client_A$, analyze the changes in ciphertext and keys to obtain gradient changes, and implement a data reconstruction attack.

However, according to our security protocol, honest $clients$ must add noise to their public datasets, making it difficult to satisfy the above conditions and significantly increasing the difficulty of attacks.

### G.2 ATTACKER-ACCESSIBLE INFORMATION AND ITS EFFECTIVENESS

In this section, we will discuss in detail the potential uses of the information obtained by attackers, and analyze the limitations of this information in implementing attacks.

1. **Information obtained after training**: The effectiveness of the information $\{\theta_{\text{Enc}}, \theta_{\text{Agg}}, \theta_{\text{Attack}_1}, \theta_{\text{Attack}_2}\}$ obtained after training is similarly limited. Through PCAPD and $\theta_{\text{Attack}}$ attack experiments, we indirectly demonstrate that there are significant differences in encryption strength and security between Private Models and Original Models. The difference in accuracy between the two models further confirms this.

2. **Information from the PPU process**: Information obtained during the PPU process can be divided into two categories:

(a) Public datasets $\{D_{\text{pub}}^i\}_{i=1}^N$: Their effectiveness is severely impacted due to the addition of noise greater than Gaussian noise. We indirectly demonstrate this through PCAPD attack experiments.

(b) Aggregation model parameters $\{\theta_{\text{Agg}}^t\}_{t=1}^T$ and their changes $\{\nabla\theta_{\text{Enc}}^t, \nabla\theta_{\text{Agg}}^t\}_{t=1}^T$: The effectiveness of this information is low because it cannot further obtain plaintext, and due to the existence of IPPU, the encryption model parameters have been further modified, intuitively reducing their usefulness again.

3. **Information obtained during the usage phase**: When there are at least two honest clients, the effectiveness of the information $\{pk_i, c_i\}_{i=1}^N, m_{\text{agg}}$ obtained during the usage phase is relatively low. Although this information can be combined with the original model to form a PCAOM attack, our experiments have demonstrated the ineffectiveness of this attack method. Intuitively, it is difficult for attackers to directly extract useful information from ciphertexts, which is similar to ciphertext-only attacks (COA) in traditional cryptography. The security of HANs is mainly based on the following two points:

(a) We adopt an encryption scheme similar to a one-time pad (OTP).

(b) The key space, plaintext space, and ciphertext space are nearly infinitely large, significantly increasing the complexity of attacks.

Based on the effectiveness analysis of all information potentially accessible to attackers, we have indirectly demonstrated the security of HANs. Through a detailed examination of information that might be leaked during the training phase, PPU process, and usage phase, we find that the effectiveness of this information in practical attacks is significantly limited. This limitation primarily stems from the design features of HANs, including the OTP encryption scheme, the vast key and ciphertext spaces, and the noise introduced in the PPU process. These factors collectively contribute to substantially increasing the difficulty of successfully implementing attacks, thereby providing robust security assurances for HANs.

---

**Algorithm 1** SampleWithReplacement

---

**Require:** Other clients' public datasets $D^{others}$, Own private data $x^{own}$, Own secret keys $sk_1$, $sk_2$
**Ensure:** Training set $T$
1: $T \leftarrow \emptyset$
2: **for** $k = 1$ to $|x^{own}|$ **do**
3:     $sample \leftarrow (sk_1[k], sk_2[k])$
4:     $y \leftarrow x^{own}[k]$
5:     **for** $D_j^{pub}$ in $D^{others}$ **do**
6:         $l \leftarrow \text{RandomInteger}(1, |D_j^{pub}|)$    ▷ Randomly select an index from the current client's dataset
7:         $(x_j^{pub}, pk_j^{pub}, c_j^{pub}) \leftarrow D_j^{pub}[l]$
8:         $sample \leftarrow sample \cup pk_j^{pub}, c_j^{pub}$
9:         $y \leftarrow y + x_j^{pub}$
10:    **end for**
11:     $sample \leftarrow sample \cup y$
12:     $T \leftarrow T \cup sample$
13: **end for**
14: **return** $T$

---

---

**Algorithm 2** CPPU

---

**Require:** Number of clients $N$, maximum iterations $maxIterations$, pre-trained HANs model (including encryption model $\theta^{enc}$ and aggregation model $\theta^{agg}$), private dataset size $privateDataSize$, public dataset size $publicDataSize$, noise scale $\sigma$

**Ensure:** Updated encryption and aggregation models for all clients

1: **for** $i = 1$ to $N$ **do**
2:     $x_i^{init}, sk_{i1}^{init}, sk_{i2}^{init} \leftarrow$ generateRandomData($publicDataSize$)
3:     $noise \leftarrow$ generateGaussianNoise($\sigma$)                         $\triangleright$ Generate initial Gaussian noise
4:     $x_i^{noisy} \leftarrow x_i^{init} + noise$                             $\triangleright$ Add initial noise to plaintext
5:     $c_i^{init} \leftarrow$ Encrypt($x_i^{init}, sk_{i1}^{init}, sk_{i2}^{init}, \theta_i^{enc}$)
6:     $D_i^{pub} \leftarrow x_i^{noisy}, (sk_{i1}^{init} + sk_{i2}^{init}), c_i^{init}$     $\triangleright$ Initialize public dataset with noisy plaintext
7: **end for**
8: **for** $t = 1$ to $maxIterations$ **do**
9:     **for** $i = 1$ to $N$ **do**
10:         $D_i^{others} \leftarrow \bigcup_{j=1, j\neq i}^{N} D_j^{pub}$              $\triangleright$ Collect all other clients' public datasets
11:         $x_i^{own}, sk_{i1}, sk_{i2} \leftarrow$ generateRandomData($privateDataSize$)   $\triangleright$ Generate private data
12:         $T_i \leftarrow$ SampleWithReplacement($D_i^{others}, x_i^{own}, sk_{i1}, sk_{i2}$)
13:         $\theta_i^{enc,new}, \theta_i^{agg,new} \leftarrow$ Optimize($\theta_i^{enc}, \theta_i^{agg}, T_i$)         $\triangleright$ Update both models
14:         $x_i^{new}, sk_{i1}^{new}, sk_{i2}^{new} \leftarrow$ generateRandomData($publicDataSize$)
15:         $c_i^{new} \leftarrow$ Encrypt($x_i^{new}, sk_{i1}^{new}, sk_{i2}^{new}, \theta_i^{enc,new}$)    $\triangleright$ Encrypt using updated encryption
16:         $noise \leftarrow$ generateGaussianNoise($\sigma$)    $\triangleright$ Generate Gaussian noise with increased scale
17:         $x_i^{noisy} \leftarrow$ **Agg**($x_i^{new} + noise$)       $\triangleright$ Apply aggregation function on noisy plaintext
18:         $D_i^{pub} \leftarrow x_i^{noisy}, (sk_{i1}^{new} + sk_{i2}^{new}), c_i^{new}$               $\triangleright$ Update public dataset
19:         $\theta_i^{enc} \leftarrow \theta_i^{enc,new}$                               $\triangleright$ Update encryption model
20:     **end for**
21:     **Broadcast** $\theta^{agg,new}$ and $D_i^{pub}$ to all clients   $\triangleright$ Synchronize the updated aggregation model and public dataset to all clients
22: **end for**

---

**Algorithm 3** IPPU

---

**Require:** Number of clients $N$, rounds per client $roundsPerClient$, HANs model after CPPU (including client-specific encryption models $\{\theta_i^{enc}\}_{i=1}^{N}$ and aggregation model $\theta^{agg}$), private dataset size $privateDataSize$, final public datasets from original CPPU $\{D_i^{pub}\}_{i=1}^{N}$

**Ensure:** Further updated encryption models for all clients

1: **for** $i = 1$ to $N$ **do**                 $\triangleright$ This loop can be executed in parallel for each client
2:     $D_i^{others} \leftarrow \bigcup_{j=1, j\neq i}^{N} D_j^{pub}$              $\triangleright$ Collect all other clients' public datasets
3:     **for** $r = 1$ to $roundsPerClient$ **do**
4:         $x_i^{own}, sk_{i1}, sk_{i2} \leftarrow$ generateRandomData($privateDataSize$)   $\triangleright$ Generate private data
5:         $T_i \leftarrow$ SampleWithReplacement($D_i^{others}, x_i^{own}, sk_{i1}, sk_{i2}$)
6:         $\theta_i^{enc} \leftarrow$ Optimize($\theta_i^{enc}, \theta^{agg}, T_i$)                    $\triangleright$ Update enc model
7:     **end for**
8: **end for**
9: **return** $\{\theta_i^{enc}\}_{i=1}^{N}$                   $\triangleright$ Return final updated encryption models

---

---

**Algorithm 4** Deep Leakage from Gradients (Zhu et al., 2019)

---

**Require:** $F(x; W)$: Differentiable machine learning model; $W$: parameter weights; $\nabla W$: gradients calculated by training data
**Ensure:** private training data $x, y$
1: **procedure** DLG($F$, $W$, $\nabla W$)
2:      $x'_1 \leftarrow \mathcal{N}(0,1), y'_1 \leftarrow \mathcal{N}(0,1)$                        ▷ Initialize dummy inputs and labels
3:      **for** $i \leftarrow 1$ to $n$ **do**
4:          $\nabla W'_i \leftarrow \frac{\partial \ell(F(x'_i, W_t), y'_i)}{\partial W_t}$                      ▷ Compute dummy gradients
5:          $D_i \leftarrow ||\nabla W'_i - \nabla W||^2$
6:          $x'_{i+1} \leftarrow x'_i - \eta \nabla_{x'_i} D_i, y'_{i+1} \leftarrow y'_i - \eta \nabla_{y'_i} D_i$       ▷ Update data to match gradients
7:      **end for**
8:      **return** $x'_{n+1}, y'_{n+1}$
9: **end procedure**

---

**Algorithm 5** DLG Attack Execution in HANs

---

**Require:** $F(x; W)$: Differentiable machine learning model; $W$: model parameters; $Enc$: encryption function; $sk_1, sk_2$: secret keys; $Attack_1, Attack_2$: adversary's attack models
**Ensure:** Attack success or failure
1: **procedure** DLG-HANs($F$, $W$, $Enc$, $sk_1$, $sk_2$, $Attack_1$, $Attack_2$)
2:      $\nabla W \leftarrow$ ComputeGradients($x, y$)                   ▷ Client computes gradients
3:      $c \leftarrow Enc(sk_1, sk_2, \nabla W)$                        ▷ Client encrypts gradients
4:      $pk \leftarrow sk_1 + sk_2$                            ▷ Public key is sum of secret keys
5:      $\nabla W_1 \leftarrow Attack_1(pk, c)$                  ▷ Attack using public key and ciphertext
6:      $\nabla W_2 \leftarrow Attack_2(c)$                       ▷ Attack using only ciphertext
7:      $(x'_1, y'_1) \leftarrow$ DLG($F, W, \nabla W_1$)            ▷ Apply DLG on first attack result
8:      $(x'_2, y'_2) \leftarrow$ DLG($F, W, \nabla W_2$)          ▷ Apply DLG on second attack result
9:      **if** $(x'_1 \simeq x \cap y'_1 \simeq y)$ or $(x'_2 \simeq x \cap y'_2 \simeq y)$ **then**
10:          **return** *Attack successful*
11:      **else**
12:          **return** *Attack failed*
13:      **end if**
14: **end procedure**

---