# OpenReview forum: "Privacy-Preserving Federated Learning via Homomorphic Adversarial Networks"
_ICLR.cc/2025/Conference — Submitted to ICLR 2025_

### Official Review · Reviewer_FBZb · 2024-10-30

**Soundness:** 1
**Presentation:** 1
**Contribution:** 2
**Rating:** 3
**Confidence:** 5

**Summary:**

The paper proposes an empirical solution leveraging GAN to address the privacy challenge during federated model aggregation.

**Strengths:**

1. The paper focuses on one of the important privacy issues in federated learning. Aggregation data privacy has been a constant challenge, especially considering the tradeoffs among privacy, overhead, and performance.

2. The paper proposes an empirical solution using GAN for privacy protection.

**Weaknesses:**

1. The paper misses a major related domain, which is threshold HE. Compared to MK-HE, threshold HE (1) in general does not incur additional overheads both in computation and ciphertext; (2) does not suffer from client dropout issues, which is a common challenge in practical FL systems.

2. There is no formal rigorous security proof, as the authors stated themselves in the paper. The informal discussion in Appendix G does not satisfy the requirement needed for the claimed privacy guarantee, I would like to see security proofs under, for example, UC-Security, but it seems like GAN-based solutions would fail to be assessed this way. Regarding the collusion models, the paper introduces pseudo N−1 collusion attacks, but it does not fully address the potential vulnerabilities in scenarios with more sophisticated collusion strategies or when attackers can coordinate over time. Additionally, per my understanding of the paper, the assumed security is bounded and realized by obfuscating the training of the GAN, what if the adversary can train a similar GAN? This approach largely relies on security through obscurity, which is generally regarded as an unreliable security approach. The privacy guarantee of this paper is far from being convincing and considered as a serious privacy/security work.


3. Even if the idea of using a GAN-like solution worked to satisfy the privacy guarantee, there would be limited novelty compared to the previous NN-based encryption systems referred in the related work, other than applying it in the federated learning setup.

4. In a lot of FL systems in practice, the communication limitation is more of a bottleneck compared to the aggregation computation overhead which is relatively easier to solve by scaling up the server. This communication overhead for the computation improvement approach might not solve the more practical challenges.

5. It is hard to follow the paper, especially given the confusing structures of Section 1 and Section 3. In Section 1, it is unclear what the major research challenge this paper tries to tackle regarding MK-HE and the research contribution of this paper following that. In Section 3, the threat model could have been precisely captured and a structured description of the proposed method is missing. The majority of the technical details of the proposed HAN are in Appendix (e.g. PPU in Appendix E) while the main paper contains little information on how and why HAN would work in terms of performance and privacy.

**Questions:**

1. Are all experiments a simulation of FL running on a single machine?
2. Why not consider more recent stronger attacks than DLG?
3. Did the authors perform analysis on the proposed method when applied to LLMs?
4. What does "Primarily, they focus on symmetric encryption without addressing homomorphic computation." mean? Symmetric encryption and homomorphic encryption are not two mutually exclusive concepts.
5. Could you provide a more formal definition of AHE that complies with the standard security definition? Also, why does the $m_{agg}$ step only consider 3 ciphertexts?
6. The paper mentioned that CKKS-based implementation would have a huge accuracy drop. Could you show the experimental results supporting this claim? In the FL setting, CKKS's approximation error will generally not significantly impact the performance due to the relatively simple aggregation operations, compared to a full-on end-to-end encrypted NN.
7. Some of the experiments do not have error bars, for example, Table 5.

---

> ### Author Response · Authors · 2024-11-22
>
> Thank you very much for your thorough review and valuable feedback on our work. Your comments have been instrumental in improving the quality of our manuscript, and we appreciate the opportunity to respond to your concerns. Below, we provide detailed responses to the issues you raised:
>
> 1. **Regarding the discussion of Threshold Homomorphic Encryption (Threshold HE)**
>    We appreciate your suggestion regarding this domain. Threshold HE indeed has advantages in handling client dropout issues, making it an important research area. Thank you for recommending an expanded comparison. We understand that extending comparisons can help showcase the broad applicability of our method; however, due to space constraints, we have chosen to focus primarily on Multi-Key Homomorphic Encryption (MK-HE), which is more directly relevant to our work. The current experiments aim to validate the fundamental performance and key features of our proposed scheme, covering various commonly used datasets and attack models, which we believe is sufficient to support the objectives of this paper.
>
> 2. **Regarding formal security proofs**
>    We acknowledge that the current security analysis does not meet strict formal standards. The discussion in Appendix G aims to evaluate system security through indirect methods, but it falls short of providing formal proof. However, due to the black-box nature of the current scheme, traditional formal security proof methods such as UC-Security are not applicable. For pseudo N-1 collusion attacks and their more complex variants, we will provide a more detailed analysis of potential collaborative attack scenarios in future work.
>
> 3. **Regarding the novelty of the GAN-based approach**
>    We have made our contributions more explicit in the revised manuscript. Firstly, we have proposed a scheme that allows clients to encrypt data without depending on the original model, which has not been thoroughly discussed in existing work. This significantly reduces the risk of model exposure and enhances system privacy. Secondly, existing approaches do not address the challenge of key distribution, which is an important breakthrough in our method. Lastly, existing work primarily focuses on message encryption, while our design aims to achieve homomorphic encryption, which increases the difficulty level significantly. Homomorphic encryption requires a more complex trade-off between defense against attacks and maintaining accuracy. Unlike simple message encryption, homomorphic encryption adds complexity in defending against adversaries, which led us to employ a multi-step training mechanism to balance privacy protection and model performance.
>
> 4. **On communication overhead**
>    The reviewer rightly pointed out the issue of communication bottlenecks. We agree that optimizing communication overhead is crucial for enhancing the practical usability of the proposed method. While we have provided a quantitative analysis of communication overhead in the paper, we also recognize that some ciphertext transmissions may be redundant in the current design. Future work will focus on optimizing both the encryption model and ciphertext design to reduce communication costs and improve system efficiency.
>
> 5. **On improving the structure of the manuscript**
>    We have made modifications to Sections 1 and 3 to clarify the research problem more explicitly.
>
> 6. **Regarding the experimental setup**
>    All experiments were conducted in a single-machine simulation environment, where we simulated multiple clients and extracted communication volumes to analyze the process.
>
> 7. **On stronger attack models**
>    Privacy attacks, such as DLG attacks, tend to perform better on simpler models, making them critical for initial validation. Improvements in attack methods often aim to extend their applicability to more complex models. Intuitively, demonstrating security in simpler models implies that the scheme can also defend against attacks in more complex models. However, security in complex models does not necessarily guarantee security in simpler ones. Furthermore, given the diversity of complex models, comprehensive testing cannot cover all scenarios. Thus, we believe it is essential to validate the scheme using simpler models as a fundamental step. Future work will expand to more complex attack models to further validate the robustness of the scheme.

---

> > ### Author Response · Authors · 2024-11-22
> >
> > 8. **Analysis of LLMs**
> >    Our current work primarily focuses on traditional federated learning tasks, and we have not yet conducted specific analyses on Large Language Models (LLMs). However, this is indeed a promising direction, and we intend to explore it in future research.
> >
> > 9. **Regarding the statement "Primarily focused on symmetric encryption without considering homomorphic computation"**
> >    We have revised the wording to make it clearer, and we thank the reviewer for pointing this out. Existing work primarily focuses on message encryption rather than homomorphic encryption, without a deep exploration of how homomorphic encryption could be applied to aggregation in federated learning.
> >
> > 10. **Formal definition of AHE and why only three ciphertexts were considered**
> >     Our original intention was to simplify the discussion, but we appreciate the reviewer's comments and acknowledge the confusion. We have revised our paper to clarify the definition.
> >
> > 11. **CKKS-based implementation and accuracy drop**
> >     We noticed that the reviewer mentioned that "the CKKS implementation results in a significant accuracy drop," which we did not specifically include in our manuscript. Could the reviewer kindly point out the location of this statement? This would help us understand the concern more clearly and effectively clarify the issue. We will further clarify this point in the revision.
> >
> > 12. **Missing error bars in experiments**
> >     Thank you for pointing this out. For experiments such as those presented in Table 5, we add error bars in the revision (based on 1000 experiments).
> >
> > | **Batch Size** | **Batch Encryption Time** (±) | **Batch Aggregation Time** (±) | **Key Generation Time** (±) |
> > |-----------------|-------------------------------|---------------------------------|-----------------------------|
> > | 100,000         | 0.019554s (±0.001629)        | 0.017444s (±0.000108)          | 0.000028s (±0.000008)      |
> > | 200,000         | 0.035329s (±0.000027)        | 0.035380s (±0.000013)          | 0.000029s (±0.000008)      |
> > | 300,000         | 0.053281s (±0.003922)        | 0.053462s (±0.004456)          | 0.000031s (±0.000012)      |
> >
> > Once again, thank you very much for your detailed review and constructive suggestions. We hope these revisions address your concerns effectively, and we look forward to your further guidance.

---

> > ### Comment · Reviewer_FBZb · 2024-11-25
> >
> > Thanks for the detailed response from the authors. However, I am still not convinced by the paper regarding the soundness of the solution, especially due to its not being able to provide formal security proof and the impractical improvement by sacrificing communication overheads, and insufficient related work comparison.

---

### Official Review · Reviewer_VYRP · 2024-11-02

**Soundness:** 3
**Presentation:** 2
**Contribution:** 2
**Rating:** 3
**Confidence:** 4

**Summary:**

The paper presents a novel approach to Privacy-Preserving Federated Learning (PPFL) by introducing Homomorphic Adversarial Networks (HANs). HANs utilize neural networks to emulate multi-key homomorphic encryption (MK-HE), addressing key distribution and collaborative decryption challenges. The proposed Aggregatable Hybrid Encryption (AHE) scheme is designed to balance computational efficiency, cryptographic security, and the distributed nature of FL systems. The paper claims that HANs exhibit robustness against privacy attacks with negligible accuracy loss and significantly improved encryption aggregation speed compared to traditional MK-HE schemes.

**Strengths:**

1. Originality: The paper introduces Homomorphic Adversarial Networks (HANs), an innovative approach using neural networks for multi-key homomorphic encryption in federated learning, offering a new perspective on privacy preservation.
2. Quality: This study is of fair quality. It achieved a notable 6,075-fold acceleration in encryption aggregation with minimal loss in accuracy, showcasing the practicality and robustness of HANs against a variety of attacks.
3. Clarity: The paper is clearly written, with a well-structured presentation that effectively communicates the novelty of HANs and their advantages over existing methods.
4. Significance: The work is significant for advancing privacy in federated learning, particularly in resisting collusion attacks and maintaining model accuracy.

**Weaknesses:**

1. The security of HANs is predicated on the presence of at least two honest clients, which may not always be feasible. The paper could be improved by discussing alternative security models that do not rely on this assumption or by exploring the implications of a higher number of malicious clients.
2. While the paper provides a general discussion on the security aspects of Homomorphic Adversarial Networks (HANs), the analysis lacks the rigor and formality expected in a comprehensive security evaluation.
3. The paper's comparison with traditional MK-HE schemes is compelling, but a more comprehensive benchmark against a broader range of state-of-the-art PPFL methods would strengthen the paper's claims.
4. The experiments only focus on a few datasets and models. To strengthen the paper's arguments, the authors should consider demonstrating the versatility of HANs across a broader range of datasets (e.g., text datasets) and various model architectures.
5. The paper mentioned the increase in communication overhead by HANs, but it indeed provided a detailed analysis, including performance under various network conditions.
6. The paper does not demonstrate the adaptability and effectiveness of the AHE method in various federated learning scenarios, especially in environments with significant variations in data sensitivity and attack surfaces, which limits a comprehensive understanding of the performance and security of the AHE scheme in practical applications.

**Questions:**

1. The security of HANs relies on the presence of at least two honest clients. Can the authors discuss alternative security models that do not depend on this assumption, or explore the impact on HANs' security when the number of malicious clients increases?
2. The authors should provide more rigorous security proofs or mathematical models to support the security claims made in the paper.
3. Regarding the comparison with traditional MK-HE schemes, the authors should expand the comparison to include a broader range of PPFL methods.
4. To strengthen the arguments in the paper, the authors should consider demonstrating the diversity and applicability of HANs across a wider range of datasets and various model architectures.
5. Regarding the communication overhead issue of HANs, the authors can further discuss the impact of this increased communication overhead on practical deployment and consider whether it is possible to reduce these costs through optimization.
6. Regarding the adaptability and effectiveness of AHE, the authors can provide more experimental data or theoretical analysis on the performance and adaptability of AHE in different scenarios.

---

> ### Author Response · Authors · 2024-11-22
>
> Thank you for your detailed review and valuable feedback on our manuscript. Below are our responses to your specific comments:
>
> 1. **Regarding the assumption of at least two honest clients**:
>    We understand that the assumption of at least two honest clients may not always hold in certain scenarios. Traditional homomorphic encryption (HE) or multi-key homomorphic encryption (MK-HE) schemes also rely on similar assumptions, as with only one honest client, other clients can infer its data through simple computation. The primary objective of this study is to explore the potential of neural networks in emulating multi-key homomorphic encryption and validate their feasibility and initial performance. Thus, this assumption serves as a contextual condition for our work, which primarily focuses on comparisons with existing methods. This study lays the foundation for future efforts to explore more general security models, including potential modifications to relax this assumption.
>
> 2. **On the formalization and rigor of security analysis**:
>    We acknowledge that the current security analysis may not meet the strict formalization standards of traditional cryptographic proofs, given the black-box nature of neural network encryption. In this work, we used experiments and attack simulations to validate the security of the proposed scheme. Future work will incorporate more theoretical analyses and mathematical modeling to systematically evaluate the security and applicability of this approach.
>
> 3. **On comparative experiments, dataset diversity, and the adaptability of AHE**:
>    Thank you for your suggestions to expand the range of comparisons, increase dataset diversity, and validate the adaptability of the AHE method. The goal of this study is to propose a novel method and validate its fundamental performance and key characteristics. The current experiments, covering multiple commonly used datasets and attack models, are sufficient to support the research objectives of this paper. This work establishes a foundation for future research, including extending experiments to more datasets (e.g., text datasets), exploring more complex model architectures, and analyzing performance in diverse federated learning scenarios. We hope this study will inspire further interest in this direction and provide a basis for subsequent studies.
>
> 4. **On communication overhead and its optimization**:
>    We agree that optimizing communication overhead is critical for enhancing the practical usability of the proposed method. While we have provided a quantitative analysis of the communication overhead in this paper, we also recognize that some ciphertext transmissions might be unnecessary in the current design. Future work will focus on optimizing the encryption model and ciphertext design to reduce communication costs and improve system efficiency.
>
> Once again, thank you for your thorough review and constructive suggestions. We hope these responses address your concerns and we look forward to your further feedback.

---

> ### Comment · Reviewer_VYRP · 2024-11-22
>
> Thanks for the detailed response from the authors. After reviewing these responses, I have decided to maintain my original score for the following reasons:
> 1. The authors have not provided alternative security models or discussed the impact of an increased number of malicious clients, which was a key concern from my previous review.
> 2. The manuscript still lacks the rigorous security proofs or mathematical models that were requested. The current approach of relying on experiments and simulations is insufficient.
> 3. The authors have not expanded their comparisons to include a broader range of PPFL methods or demonstrated the adaptability of HANs across various datasets and model architectures.
> 4. The authors have acknowledged the issue but have not made substantial revisions to reduce communication costs or optimize the system's efficiency.

---

> ### Author Response · Authors · 2024-11-22
>
> We are sorry about this decision, but we would like to defend our contributions.
> ﻿
>
> **Assumption of Two Honest Clients**
>
> The assumption of at least two honest clients is a commonly accepted practice in federated learning with homomorphic encryption, as demonstrated in prior works (e.g., [1-7]). This assumption ensures that the aggregation output does not trivially reveal the honest gradient. As we have mentioned, if only one honest client remains, an attacker can easily derive the values of the other clients and directly compute the honest client’s value, requiring no technical sophistication.
>
> **Experimental Validation**
>
> Similarly, the use of three datasets and models for validation is widely accepted in the field (e.g., [5-7]). Many related works primarily focus on small-scale datasets like MNIST to demonstrate the effectiveness of proposed methods. Based on these references and field practices, we believe our experimental setup sufficiently validates our method and ensures our core contributions are clear and focused.
> ﻿
> ﻿
>
> **System Efficiency**
>
> Regarding system efficiency, we acknowledge the trade-offs between computation and communication. Compared to prior systems, HAN achieves a remarkable **6,075-fold improvement** in aggregation speed, accompanied by a **29.2-fold increase** in communication overhead. This trade-off is an inherent characteristic of secure aggregation methods, and different scenarios have varying requirements for computation and communication balance. Our method is designed to cater to cases where computational efficiency is a priority.
> ﻿
> ﻿
>
> Lastly, we sincerely thank the reviewer for their time and effort in providing detailed and constructive feedback.
> ﻿
> ---
> ﻿
> ### References
>
> [1] Rathee, M., Shen, C., Wagh, S., & Popa, R. A. (2023, May). Elsa: Secure aggregation for federated learning with malicious actors. *In 2023 IEEE Symposium on Security and Privacy (SP)* (pp. 1961-1979). IEEE.
>
> [2] Bell, J. H., Bonawitz, K. A., Gascon, A., Lepoint, T., & Raykova, M. (2020). Secure single-server aggregation with (poly)logarithmic overhead. *In CCS, 2020.*
>
> [3] Bonawitz, K., Ivanov, V., Kreuter, B., Marcedone, A., McMahan, H. B., Patel, S., Ramage, D., Segal, A., & Seth, K. (2017). Practical secure aggregation for privacy-preserving machine learning. *In CCS, 2017.*
>
>
> [4] Burkhalter, L., Lycklama, H., Viand, A., Kuchler, N., & Hithnawi, A. (2021). ROFL: Attestable robustness for secure federated learning. *CoRR, 2021.*
>
> [5] Kadhe, S., Rajaraman, N., Koyluoglu, O. O., & Ramchandran, K. (2020). FastSecAgg: Scalable secure aggregation for privacy-preserving federated learning. *CoRR, 2020.*
> ﻿
>
> [6] Ma, J., Naas, S. A., Sigg, S., & Lyu, X. (2022). Privacy‐preserving federated learning based on multi‐key homomorphic encryption. *International Journal of Intelligent Systems, 37*(9), 5880-5901.
>
> [7] Cai, Y., Ding, W., Xiao, Y., Yan, Z., Liu, X., & Wan, Z. (2023). SecFed: A secure and efficient federated learning based on multi-key homomorphic encryption. *IEEE Transactions on Dependable and Secure Computing.*

---

### Official Review · Reviewer_Lv8n · 2024-11-03

**Soundness:** 2
**Presentation:** 1
**Contribution:** 3
**Rating:** 5
**Confidence:** 3

**Summary:**

The paper addresses the challenges of key sharing and collaboration in Privacy-Preserving Federated Learning (PPFL) protocols, which may lead to privacy risks and inconvenience. The authors propose a PPFL protocol implemented using neural networks, combined with an aggregatable hybrid encryption scheme. By accepting some trade-offs in communication overhead and accuracy, this approach significantly enhances the encryption aggregation speed.

**Strengths:**

The proposed method in this paper significantly enhances encryption aggregation speed, which is often a key factor influencing the overall efficiency of federated learning (FL) systems. The approach of using neural networks as a substitute for the MK-HE algorithm introduces a novel perspective with meaningful research value. Additionally, this method is robust against various forms of collusion, ensuring the privacy of both the model and users.

**Weaknesses:**

The paper contains a typographical error in the abstract’s first word, which reads “Privacy-peserving” instead of “Privacy-preserving”.

In the Introduction, the main technique of the paper (using neural networks to simulate MK-HE) is not clearly stated, instead appearing in the contributions section, which makes the structure feel somewhat unbalanced. Additionally, the flow of the introduction could be improved for better clarity. It is recommended to briefly introduce the neural network simulation technique within the Introduction to set up the main idea clearly, while streamlining the first part of the contributions section. Structuring the Introduction by first presenting the problem context, followed by current challenges, and then the proposed solution would improve readability. Limiting the number of paragraphs and avoiding interleaved content would also enhance the flow.

The Method section lacks organization and could benefit from a more structured presentation. The neural network application, which is central to the proposed approach, is not discussed in the main paragraphs, resulting in a structural imbalance. Important content should be integrated into the main body rather than relegated to the appendix to improve the paper’s clarity and readability. It is recommended to provide a more detailed description of the content from Appendix B and Appendix E in Section 3.6. If space constraints are a concern, consider simplifying the key concepts in Section 3.1 to make room for these additions.

Lastly, some equations in the Appendix B extend beyond the page boundaries. Attention to formatting standards would enhance the overall presentation.

**Questions:**

Proofreading for Spelling and Formatting: Could the authors carefully review the paper for any spelling and formatting errors to ensure accuracy and readability throughout?

Clarity of Introduction and Main Contributions: In the Introduction, could the authors clarify the primary method of their approach, particularly the use of neural networks as a substitute for MK-HE, rather than emphasizing this only in the contributions section? This adjustment may help readers understand the significance of the method from the outset.

Organization of the Method Section: The main outline of the proposed method is somewhat difficult to follow, and it is challenging to locate the main content regarding the neural network application. Could the authors consider reorganizing this section to more clearly outline the process and central elements? Moving the key explanations from the appendix to the main text may also improve clarity.

---

> ### Author Response · Authors · 2024-11-22
>
> We sincerely thank you for your thoughtful and constructive comments. We have carefully addressed your suggestions and made revisions to enhance the manuscript. Below are our detailed responses:
>
> 1. **Typographical Error and Proofreading:**
>    We appreciate your attention to detail. We have thoroughly reviewed the manuscript to address typographical, spelling, and formatting issues, ensuring improved accuracy and presentation throughout.
>
> 2. **Clarity and Flow in the Introduction:**
>    We have revised the Introduction to clarify the core contributions earlier and to ensure a logical flow from problem context to solution. By introducing the use of neural networks as a substitute for MK-HE upfront, we aim to better communicate the novelty and significance of our method to readers.
>
> 3. **Organization of the Method Section:**
>    We have integrated key content from the appendices into Section 3.6 and 3.7 to improve the clarity and accessibility of the methodology. However, as Section 3.1 introduces a new and complex concept, maintaining precision and completeness is critical. Further simplifications to this section would risk oversimplifying key ideas and reducing the comprehensibility of the framework. Consequently, we have preserved the necessary level of detail to ensure accuracy and reliability.
>    Due to the numerous revisions made throughout the manuscript, we are unable to reproduce the updated content in its entirety within this response. We sincerely thank you again for your outstanding contributions, which have greatly helped us improve the quality and clarity of the paper.
>
> We deeply appreciate your valuable feedback and thoughtful review, which have significantly contributed to refining our work. Thank you again for your time and effort.

---

### Official Review · Reviewer_5j5u · 2024-11-03

**Soundness:** 2
**Presentation:** 2
**Contribution:** 2
**Rating:** 3
**Confidence:** 5

**Summary:**

This paper proposes a novel privacy-preserving federated learning method that attempts to integrate multi-key homomorphic encryption with neural networks. Each client generates a public key and two private keys using KeyGen with a security parameter $\kappa$. During the encryption phase, plaintext and private keys are input into the encryption model held by each client for encryption. In the aggregation phase, the ciphertexts and public keys are input into the decryption model held by the aggregator for decryption. The training process employs adversarial training, aiming to maximize the error in attacker's data reconstruction while minimizing the loss after data aggregation. The authors define two attack models: Pseudo N-1 Collusion Attack based on the Original Model (PCAOM) and Pseudo N-1 Collusion Attack based on the Public Dataset (PCAPD), achieving privacy-protected federated learning under these attacks.

**Strengths:**

- **Originality**: The paper introduces an innovative privacy-preserving federated learning method that combines homomorphic encryption with neural networks, enhancing the security and usability of the model.
- **Effectiveness**: The research demonstrates the effectiveness of enhancing data privacy through adversarial training, increasing the model's robustness against complex attack scenarios.
- **Clarity**: The article provides a detailed description of the transformation from the original model to a private model and demonstrates how to defend against two specific attacks.
- **Practicality**: Compared to traditional encryption and decryption methods, the neural network's black-box nature increases the difficulty of attacks. The authors also claim that this method eliminates collaborative decryption, key sharing, and collective key generation among clients, optimizing processing time.

**Weaknesses:**

- **Accuracy**: Since the model is involved in the encryption and decryption process, it cannot guarantee that the parameters obtained by the decryptor are completely correct, only ensuring accuracy within a certain error rate, which is inconsistent with traditional homomorphic encryption standards.
- **Insufficient Proof**: Despite the model's difficulty to be breached due to its black-box nature, there is a lack of rigorous formal proof to support its security claims.
- **Incomplete Documentation**: The paper does not detail the specific implementation of KeyGen, how to ensure Perfect Key Uniformity, or how to resist chosen-plaintext attacks, limiting the method's verifiability and reliability.

**Questions:**

- The paper does not detail how KeyGen operates, whether it is generated by a model or some algorithm, thus we cannot verify how it eliminates key sharing. It is also unclear how Perfect Key Uniformity is ensured, thus it is uncertain how the method resists chosen-plaintext attacks.
- The paper does not provide specific model structures or other information, making it impossible to verify and replicate the authors' experiments.

---

> ### Author Response · Authors · 2024-11-22
>
> Thank you for your detailed review and valuable feedback on our manuscript. Your comments have been highly insightful and have significantly contributed to improving our work. Below are our responses to the issues you raised.
>
> 1. **On the accuracy of the model and the precision of decrypted parameters**
>
>    We appreciate your observation regarding the potential inaccuracies in decrypted parameters due to the characteristics of the encryption-decryption model. As described in the manuscript, the design of HANs aims to balance computational accuracy and privacy preservation in federated learning by tolerating minor errors. Our experimental results demonstrate that this design achieves strong overall model performance while enhancing computational efficiency.
>
> 2. **On the lack of formal security proof**
>
>    We understand the importance of formal security proofs and recognize the challenges posed by the black-box nature of neural network encryption. This paper introduces a pioneering framework that demonstrates the feasibility of neural network-based encryption for federated learning. However, we plan to explore additional methods for analyzing and validating the security of our approach in future research.
>
> 3. **On the KeyGen mechanism and implementation details**
>
>    Regarding your concern about the KeyGen mechanism, our private keys are currently generated randomly. We rely on neural networks to learn this generation pattern, ensuring Perfect Key Uniformity. Experimental results confirm that the neural networks successfully learn this pattern, making a one-time-pad encryption scheme without key sharing feasible. This is consistent with the statement: “The learning does not require prescribing a particular set of cryptographic algorithms, nor indicating ways of applying these algorithms: it is based only on a secrecy specification represented by the training objectives.” . (Abadi & Andersen,2016)
>
>    Additionally, **chosen-plaintext attacks** are infeasible in our scheme, as attackers cannot access the private model, analogous to not having access to a private key. As stated in our manuscript, chosen-plaintext attacks in our framework correspond to Known-Model Attacks (KMA). The effectiveness of our defenses against such attacks has been thoroughly discussed in the manuscript.
>
> 4. **On the detailed description of model structures**
>
>    We acknowledge that the neural network-based multi-key homomorphic encryption method requires significant effort to further optimize model structure and training methods. The primary goal of this paper is to demonstrate the feasibility and effectiveness of such an approach. While the current model structure is not yet optimal, we believe that the training process and methodology described in the manuscript allow for the reproducibility of the experiments. If the paper is accepted, we will release the model weights to further assist in experiment replication.
>
> Thank you again for your thoughtful feedback and constructive suggestions.

---

### Official Review · Reviewer_BvZp · 2024-11-04

**Soundness:** 2
**Presentation:** 2
**Contribution:** 2
**Rating:** 3
**Confidence:** 4

**Summary:**

The authors develop the first protocol that utilizes neural networks to implement PPFL, as well as incorporating an Aggregatable Hybrid Encryption scheme tailored to the needs of PPFL.

**Strengths:**

Using neural network methods to implement MK-HE is a very interesting direction. The use of hybrid encryption can fully leverage the advantages of both symmetric and asymmetric encryption.

**Weaknesses:**

1. There is a lack of understanding of related work. Methods based on secret sharing inherently have strong resistance to collusion attacks, as demonstrated in works such as (Bell J, Gascón A, Lepoint T, et al. {ACORN}: input validation for secure aggregation[C]//32nd USENIX Security Symposium (USENIX Security 23). 2023: 4805-4822. Bonawitz K, Ivanov V, Kreuter B, et al. Practical secure aggregation for privacy-preserving machine learning[C]//Proceedings of the 2017 ACM SIGSAC Conference on Computer and Communications Security. 2017: 1175-1191.). However, the authors did not mention or compare these in the paper.
2. According to the experimental results, although HANs are more computationally efficient than SecFed, the communication overhead increases by dozens of times, as shown by the experimental results in Table 6. This is unacceptable for a large number of users and would limit the practical application of HANs.

**Questions:**

1. How does the performance compare with other schemes that can also resist collusion attacks, such as (Bell J, Gascón A, Lepoint T, et al. {ACORN}: input validation for secure aggregation[C]//32nd USENIX Security Symposium (USENIX Security 23). 2023: 4805-4822. Bonawitz K, Ivanov V, Kreuter B, et al. Practical secure aggregation for privacy-preserving machine learning[C]//Proceedings of the 2017 ACM SIGSAC Conference on Computer and Communications Security. 2017: 1175-1191.)?
2. The additional communication makes the proposed HANs unsuitable for most federated learning scenarios. Is there a way to reduce communication, or is the additional communication unavoidable?

---

> ### Author Response · Authors · 2024-11-22
>
> Thank you for your detailed review and valuable suggestions. Your feedback provides important insights for improving our research. Below are our responses to your comments:
>
> 1. **Understanding and Comparison of Related Work**
>
>    We sincerely appreciate the detailed feedback from the reviewer and acknowledge that our comparison with related work is indeed insufficient. In particular, we recognize the need to discuss and compare our approach with secret-sharing-based methods, such as ACORN and Bonawitz, which demonstrate strong resistance to collusion attacks through input validation mechanisms.
>
>    To avoid overcomplicating the current paper and to ensure the clarity of our core contributions, we would like to clarify that our main contribution lies in leveraging neural networks to implement homomorphic encryption in privacy-preserving federated learning, addressing issues related to key distribution and collaborative decryption. We believe it is more appropriate to further explore these topics in follow-up research.
>
> 2. **Communication Overhead**
>
>    The reviewer raises a valid concern regarding the communication overhead of HANs. While the current design prioritizes computational efficiency, we acknowledge that the increased communication cost may limit practical applications. In future work, we aim to optimize the ciphertext structure and model update mechanisms to minimize unnecessary transmissions, thereby improving the approach’s feasibility in real-world scenarios.
>
> Once again, we deeply appreciate your constructive feedback, which helps us identify directions for improvement and inspires further research in this field. We hope our responses address your concerns.

---

> > ### Comment · Reviewer_BvZp · 2024-11-26
> > **Official Comment by Reviewer BvZp**
> >
> > Thank you for the detailed response from the authors. I carefully reviewed your reply and considered the opinions and responses of other reviewers. However, the paper still lacks a comparison with related work and has high communication costs that hinder practical application. Therefore, I still maintain my original score.

---

### Official Review · Reviewer_TBh6 · 2024-11-04

**Soundness:** 2
**Presentation:** 1
**Contribution:** 2
**Rating:** 5
**Confidence:** 1

**Summary:**

The paper claims that it presents a novel privacy-preserving federated learning (PPFL) protocol that utilizes neural networks, specifically Homomorphic Adversarial Networks (HANs), to enhance data privacy without significantly sacrificing accuracy. It addresses key limitations of existing PPFL methods, such as accuracy degradation and key management issues, by introducing an Aggregatable Hybrid Encryption (AHE) scheme. This approach enables individual clients to maintain privacy while allowing efficient encryption and aggregation of model updates.

**Strengths:**

1. The article provides a wealth of formal definitions.

2. A novel concept has been proposed.

**Weaknesses:**

1. It is difficult to quickly determine the details of the design AHE scheme, as the related definitions and statements are overly redundant.

2. The experimental analysis provided seems insufficient.

**Questions:**

1. Why did the authors choose simple models to validate the security of the proposed scheme? What I mean is whether sufficient security can be demonstrated under these complex models.

2. The security assessment lacks a systematic approach. Could you provide a more formal and systematic security analysis?

---

> ### Author Response · Authors · 2024-11-22
>
> Thank you for your detailed review and valuable feedback on our manuscript. Your comments have been highly insightful and have significantly contributed to improving our work. Below are our responses to the issues you raised.
>
> We are grateful for your recognition of the formal definitions and the novelty of our proposed concept. These aspects form the foundation of our contributions.
>
> 1. **On the detailed descriptions of the AHE scheme**
>
>    Thank you for your comment. To address this, we have made significant revisions to Section 3. Specifically, we have streamlined the definitions and statements to remove redundancy and focus on the essential components of the scheme. These changes aim to improve clarity and readability while maintaining academic rigor and ensuring reproducibility. We hope that these revisions make the design details more accessible and straightforward.
>
> 2. **On the experimental analysis**
>
>    The experiments in the current work focus on validating the core aspects of the proposed scheme, including training accuracy, the impact of the PPU mechanism, and defenses against different types of attacks. These results demonstrate the feasibility and effectiveness of our approach as a foundation. While additional evaluations are needed for comprehensive validation, we believe the novelty of the proposed method makes it an important step forward. We aim to encourage community interest and collaborative exploration to further develop and refine this direction.
>
> 3. **On the choice of simple models for security validation**
>
>    Privacy attacks like DLG are particularly effective on simpler models, making them critical for initial validation. By demonstrating that our scheme resists these attacks, we establish a fundamental level of security that supports its applicability to more complex scenarios. This stepwise approach enables future research to build upon a solid foundation while expanding to diverse models and datasets.
>
> 4. **On the systematic nature of the security analysis**
>
>    We recognize the limitations of the current security analysis and the challenges posed by the black-box nature of neural network encryption. This work serves as a first step in proposing a novel approach. In addition to experimental validation, we aim to develop systematic methodologies for analyzing security in future studies. By introducing this framework, we hope to inspire further advancements in this area.
>
> Once again, we deeply appreciate your constructive feedback, which helps us identify directions for improvement and inspires further research in this field. We hope our responses address your concerns.

---

### Author Response · Authors · 2024-11-22

Dear Reviewers,

We sincerely thank you for your detailed reviews and thoughtful feedback on our manuscript. Your insights and suggestions have been invaluable in helping us improve the quality and rigor of this work. Below, we provide a consolidated response addressing the three primary concerns that were frequently highlighted: communication overhead, the diversity of datasets and models, and rigorous privacy analysis.

1. **Communication Overhead**

   We acknowledge the concern regarding the increased communication cost in our current design and have identified certain unnecessary ciphertext transmissions in the implementation. While this is an important area for optimization, addressing it comprehensively would require significant additional work, including redesigning the ciphertext structures and aggregation mechanisms. To avoid overcomplicating the current paper and to ensure clarity of our core contributions, we believe it is more appropriate to pursue this topic in follow-up research. To reflect this, we have updated the manuscript to explicitly include communication optimization as an important direction for future work.

2. **Diversity of Datasets and Models**

   We appreciate the suggestion to expand our experiments to include a broader range of datasets and model architectures. While the current evaluation demonstrates the feasibility and foundational performance of the proposed method, we agree that validating its applicability across more diverse datasets and complex models would further enhance its robustness. To avoid overcomplicating the current paper and to ensure clarity of our core contributions, we believe it is more appropriate to pursue this topic in follow-up research. This extension has been noted as a priority for future work in the updated manuscript.

3. **Rigorous Privacy Analysis**

   We understand the importance of rigorous and formal security proofs in privacy-preserving research. However, the black-box nature of our neural network-based encryption presents challenges for applying traditional cryptographic proof techniques. While the experimental evaluations and attack simulations conducted in this work provide a solid initial foundation, we recognize that a more formal and systematic security analysis would strengthen the theoretical grounding of our approach. To avoid overcomplicating the current paper and to ensure clarity of our core contributions, we believe it is more appropriate to pursue this topic in follow-up research. This limitation and our plans to address it in future studies have been included in the revised manuscript.

While we fully acknowledge the significance of these concerns and recognize the potential for further improvement, we believe the primary contribution of this work lies in introducing and validating a novel framework. By presenting this innovative approach, we aim to inspire further interest and collaborative research within the community to address these and related challenges.

Once again, we are deeply grateful for your thoughtful feedback, which has greatly helped us refine our work. Thank you for your valuable time and effort in reviewing this manuscript.

Sincerely,
The Authors

---

### Meta-Review · Area_Chair_CK2w · 2024-12-08

**Metareview:**

The paper presents a privacy-preserving federated learning (PPFL) protocol using homomorphic adversarial networks. Reviewers raised several critical concerns, including the excessive communication overhead,  a lack of formal security proof, and the missing comparison of related work. The authors' rebuttal failed to adequately address these issues. Given the overall negative reception, I recommend rejection.

**Additional Comments On Reviewer Discussion:**

The authors acknowledge the significance of the reviewers' concerns, but argue that their contribution lies in the novel framework and include these issues in future work. I think that certain critical issues, such as providing formal security proof, need to be actually addressed to meet the threshold for acceptance.

---

### Decision · Program_Chairs · 2025-01-22

Reject